# HSCodeComp: A Realistic and Expert-level Benchmark for Deep Search Agents in Hierarchical Rule Application

## Abstract

Effective deep search agents must not only access open-domain and domain-specific knowledge but also apply complex rules—such as legal clauses, medical manuals and tariff rules. These rules often feature vague boundaries and implicit logic relationships, making precise application challenging for agents. However, this critical capability is largely overlooked by current agent benchmarks. To fill this gap, we introduce HSCodeComp, the first realistic, expert-level e-commerce benchmark designed to evaluate deep search agents in hierarchical rule application. In this task, the deep reasoning process of agents is guided by these rules to predict 10-digit Harmonized System Code (HSCode) of products with noisy but realistic descriptions. These codes, established by the World Customs Organization, are vital for global supply chain efficiency. Built from real-world data collected from large-scale e-commerce platforms, our proposed HSCodeComp comprises 632 product entries spanning diverse product categories, with these HSCodes annotated by several human experts. Extensive experimental results on several state-of-the-art LLMs, open-source, and closed-source agents reveal a huge performance gap: best agent achieves only 46.8% 10-digit accuracy, far below human experts at 95.0%. Besides, detailed analysis demonstrates the challenges of hierarchical rule application, and test-time scaling fails to improve performance further. Codes and the benchmark will be publicly released.

## 1 Introduction

Deep search agents have demonstrated significant value in solving complex real-world problems, where robust external knowledge utilization constitutes a critical capability (Wu et al., 2025; Tao et al., 2025; Li et al., 2025b). To evaluate this capability, numerous established benchmarks are proposed to assess agents in utilizing open-domain data (*e.g.,* GAIA (Mialon et al., 2023b) and BrowseComp (Wei et al., 2025)) and domain-specific data (*e.g.,* WebMall (Peeters et al., 2025a), FinSearchComp (Hu et al., 2025a) and MedBrowseComp (Yu et al., 2025b)).

Beyond open-domain and domain-specific data, agents also need to effectively apply rules that encode human expert knowledge, particularly in scenarios like law, medical and e-commerce (Li et al., 2025a; Chen et al., 2025; Yao et al., 2022). For instance, legal case adjudication require interpreting abstract legal provisions, and accurate e-commerce product classification in depends on tariff rules (Grainger, 2024). Previous works have defined rule application as using specific logical rules with supporting facts to derive conclusions (Wang et al., 2024; Servantez et al., 2024). In contrast, we define it as a core capability for deep search agents, where human-written rules are systematically applied to guide complex reasoning and decision-making (Sadowski & Chudziak, 2025).

Building on this observation, we categorize knowledge data for deep search agents into three levels (Figure 1, left), with increasing knowledge complexity: (1) *Level 1: Open-domain Data* - Tests understanding and deep reasoning abilities of agents on long-form web content. Established benchmarks include GAIA (Mialon et al., 2023b) and BrowseComp (Wei et al., 2025); (2) *Level 2: Structured Data* - Assesses agents to precisely utilize structured data such as databases and knowledge graphs, as seen in domain-specific benchmarks like WebMall (Peeters et al., 2025a), MedBrowseComp (Chen et al., 2025b) and FinSearchComp (Hu et al., 2025a); (3) *Level 3: Rule Data* - Evaluates

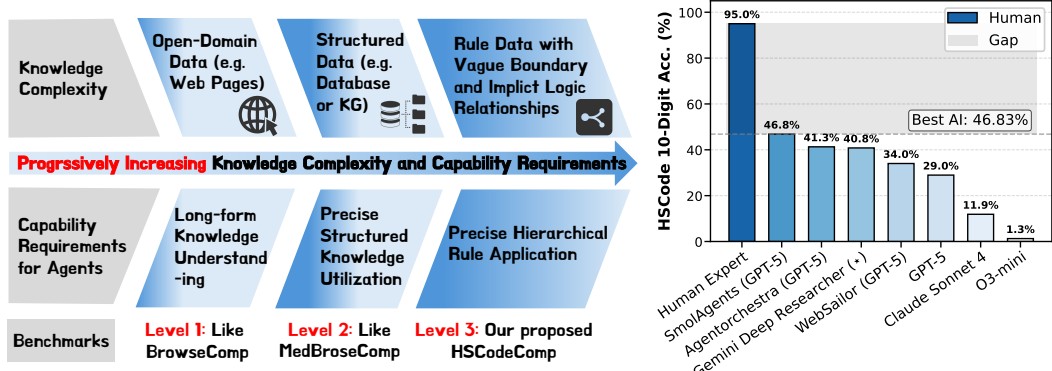

Figure 1. **Left:** Recent benchmarks reveal the increasing knowledge complexity and capability requirements for agents. **Right:** 10-digit HSCode accuracy of state-of-the-art baseline largely lags behind human experts ($46.8\% < 95.0\%$), proving the challenges of hierarchical rule application. The closed-source agent ($\star$) is evaluated on the subset due to API unavailability.

agents to apply complex and abstract rules. This level presents two key challenges: (a) making accurate decisions when rules contain vague natural language descriptions (Sadowski & Chudziak, 2025); and (b) reasoning about logical dependencies among rules, such as exception clauses and cross-category relationships (Guha et al., 2023). Despite the importance of rule application in real-world scenarios, current agent benchmarks largely overlook its evaluation.

To fill this gap, we introduce HSCODECOMP (short for the **H**armonized **S**ystem **Code** (HSCode) **Comp**etition), the first realistic, expert-level e-commerce benchmark designed to evaluate agents in predicting complete 10-digit Harmonized System Code (HSCode) of the product, using hierarchical rules (*e.g.,* eWTP tariff rules[1]). HSCodes organize products through a hierarchical structure spanning over 5,000 distinct codes across multiple classification levels, representing the global standard for classifying traded international goods, established by the World Customs Organization and implemented across more than 200 countries for customs clearance and tariff determination (Grainger, 2024; Nath et al., 2025). Built from the data of the large-scale e-commerce platforms, our proposed HSCODECOMP comprises 632 carefully curated product entries, encompassing 27 unique HS chapters and 32 distinct first-level categories. These HSCodes have been rigorously annotated by multiple e-commerce domain experts, ensuring that HSCODECOMP is expert-level. Accurately predicting the exact 10-digit HSCode presents significant challenges: agents must perform multi-hop hierarchical reasoning with complex tariff rules while processing noisy but realistic product descriptions that often contain abbreviations, language variations, or incomplete information.

Extensive experiments on the state-of-the-art baselines, including 14 advanced foundation models, 6 advanced open-source agent systems and 3 closed-source agent systems, demonstrate that HSCode prediction task remains a substantial challenge for current AI approaches. As shown in the Figure 1 (right), even the best-performing system (SmolAgent (Roucher et al., 2025) with GPT-5) achieves only 46.8% accuracy, substantially below the 95.0% accuracy attained by human experts. Further detailed analysis reveals that existing agent systems lack critical capabilities required for this complex hierarchical rule applications. Notably, test-time scaling approach—which has proven effective in other reasoning tasks (Guo et al., 2025; Liu et al., 2025)—fail to improve performance on HSCODE-COMP. These observations demonstrate the challenging nature of our proposed HSCODECOMP, highlighting the need for more effective designs of agent systems. To facilitate future research, we will publicly release codes and the benchmark dataset of HSCODECOMP.

## 2 RELATED WORKS

### 2.1 PREVIOUS WORKS IN HSCODE PREDICTION

Previous works treat HSCode prediction as the e-commerce text classification task (Grainger, 2024), using pre-trained BERT models (Liao et al., 2024; Shubham et al., 2022) or Large Language Models

---

[1] https://www.ewtp.com/web/smart/hscode

(LLMs) prompting (Gholamian et al., 2024). However, these approaches fail to leverage domain-specific knowledge, especially the rules written by human experts (Gholamian et al., 2024; Judy, 2024). Besides, existing HSCode benchmarks face two critical limitations (Judy, 2024; Lee et al., 2024; Stassin et al., 2023): (1) they are typically constructed from publicly accessible customs rulings, suffering from data leakage; (2) they are not released. In contrast, our released HSCODECOMP is collected from large-scale online shopping platforms with noisy product descriptions, making it more challenging and realistic.

## 2.2 BENCHMARKING LEVEL 1 KNOWLEDGE UTILIZATION

Numerous benchmarks have been proposed to evaluate agent capabilities in understanding and deep reasoning over long-form open-domain web content (Thomas et al., 2025; Yao et al., 2024; Joshi et al., 2017; Phan et al., 2025). For example, WebArena (Zhou et al., 2023) provides realistic, self-hostable websites with standardized evaluation protocols to assess functional correctness. Web-Shop (Yao et al., 2022) and ALFWorld (Shridhar et al., 2021) evaluate long-horizon decision-making abilities of agents in web environments through tool interactions. More recent deep search benchmarks, such as GAIA (Mialon et al., 2023a), BrowseComp (Wei et al., 2025), WebWalkerQA (Wu et al., 2025) and BrowseComp-ZH (Zhou et al., 2025), demand advanced tool-usage and deep reasoning capabilities (Zhang et al., 2025; Li et al., 2025b).

## 2.3 BENCHMARKING LEVEL 2 KNOWLEDGE UTILIZATION

Recent works have focused on how agents utilize structured knowledge in domain-specific applications. Unlike open-domain data, domain-specific knowledge is typically organized into structured formats such as databases and knowledge graphs (Huang et al., 2025; Yu et al., 2025a; Chen et al., 2025a), enabling more precise knowledge retrieval and utilization. To evaluate these capabilities, numerous deep search benchmarks have been proposed, including WebMall (Peeters et al., 2025b) and DeepShop (Lyu et al., 2025) for e-commerce, LegalAgentBench for law (Li et al., 2025a), Fin-SearchComp for finance (Hu et al., 2025a), DAgent for data analysis (Xu et al., 2025), CRMArena for CRM workflows (Huang et al., 2025), and MedBrowseComp for medicine (Chen et al., 2025b).

In summary, while there exists numerous evaluation benchmarks for assessing agent performance in open-domain or domain-specific scenarios, none evaluates the ability to apply Level 3 abstract rule-based knowledge. To address this critical gap, we introduce a realistic and expert-level e-commerce benchmark HSCODECOMP. Our benchmark presents significant challenges even for state-of-the-art closed-source and open-source agent systems.

## 3 TASK FORMULATION OF HSCODE PREDICTION

The HSCode prediction task is to assign a valid and unique 10-digit Harmonized System (HS) code to a given noisy but realistic product description. The product HSCode plays the crucial role in e-commerce system. It is the global standard for classifying traded goods, essential for tariffs, customs, and trade governance. The core challenge is to learn a mapping function, *i.e.,* agents implemented by Large Language Models (LLMs) or Vision Language Models (VLMs), $f : \mathcal{X} \rightarrow \mathcal{Y}$.

**Input:** Figure 2 shows that each product $x \in \mathcal{X}$ encompass rich records: $x = (t, A, c, i, p, u, r)$, where $t$ is the product title; $A = \{(k_j, v_j)\}_{j=1}^{K}$ represents a set of $K$ product attributes (*e.g.,* material and package size); $c$ represents product categories defined by the e-commerce platform; $i$ is the product image; $p$ and $u$ are the price and currency; and $r$ is the webpage URL of the product.

**Hierarchical Rule Utilization:** Accurate HSCode prediction requires effectively utilizing three types of e-commerce knowledge: (1) **Hierarchical tariff rules** from official platform (*e.g.,* eWTP) aggregating authoritative tariff systems including the US, which are organize in a hierarchical structure. As shown in Figure 11, these rules contain complex implicit logic relationships, for example, the exception clause in tariff rules like *excluding articles of HS heading 8539 ...* (highlighted with red boxes). Besides, these rules often employ vague linguistic constraints (highlighted with blue boxes) that challenge existing AI agents; (2) **Human-written decision rules** that specify how to correctly apply tariff rules. These rules provide high-level decision principles (see Figure F.1 for an example

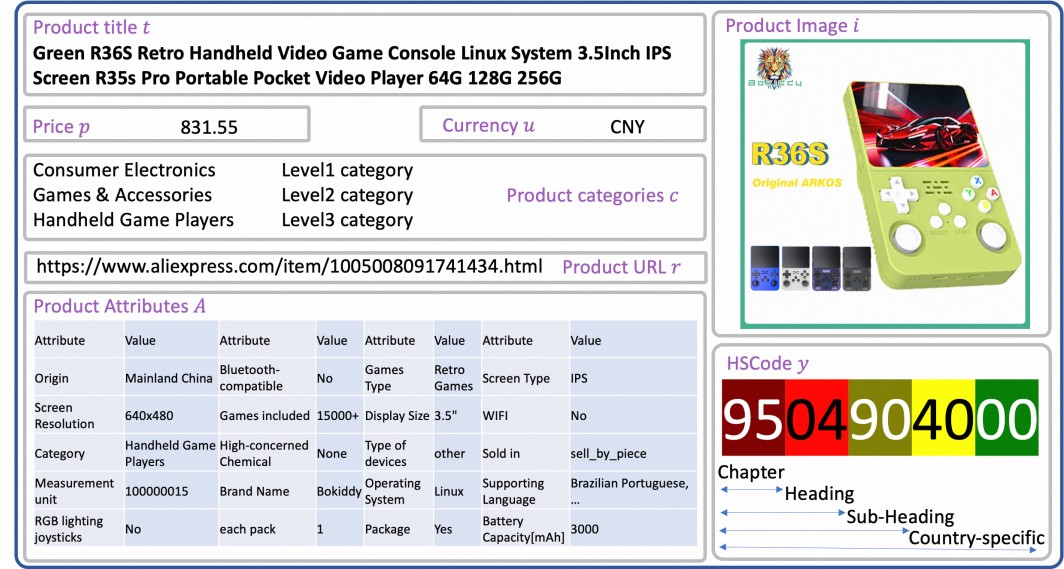

Figure 2. One example of a game console product in HSCODECOMP.

with six key principles defined by domain experts); and (3) **Official customs rulings databases**, such as the U.S. Customs Rulings Online Search System (CROSS)[2], which document historical HSCode classification decisions. As illustrated in Figure 12, these databases contain complex information format requiring advanced reasoning capabilities. All these resources in HSCODECOMP target US tariff classification, ensuring the alignment among these knowledge.

**Output:** The HSCode $y \in \mathcal{Y}$ is a single 10-digit numeric string $\mathcal{Y} \subseteq \{0, 1, \ldots, 9\}^{10}$. The HSCode is hierarchical, where first 2-digit, 4-digit, and 6-digit represents the HS chapter, heading and sub-heading of tariff classification of products, respectively, and last 4 digits (from 7 to 10) are country-specific codes. In summary, this 10-digit HSCode follows a valid path in the official HS taxonomy.

## 4 HSCODECOMP CONSTRUCTION AND EVALUATION METRICS

### 4.1 BENCHMARK CONSTRUCTION

We design a rigorous pipeline, ensuring the dataset is diverse, realistic and expert-level: (1) Data Collection and Diversity Control; (2) Human Expert Annotation; and (3) Human Expert Validation.

**Data Collection and Diversity Control.** Products in our proposed HSCODECOMP is sourced from a large-scale global e-commerce platform. These product profiles include the noisy information, ensuring that task instances reflect the real-world challenges. Besides, we also balance the data category distribution to prevent the potential topical skew. Specifically, we apply a pre-processing step: a semantic redundancy filter discards the products ($x$) sharing identical categories ($c$) and 10-digit HSCode ($y$) with existing product instances. This ensures that HSCODECOMP is not dominated by common and easy-to-classify products.

**Human Expert Annotation.** To ensure the quality of HSCODECOMP, we engage human experts specialized in HSCode classification to annotate the HSCode ($y$) for each product profile ($x$). As shown in Figure 3 (left), the annotation process follows a five-step pipeline: (1) Two experts gather comprehensive information from the product webpage (Step 1); (2) They extract the core structured features of products (Step 2); (3) Experts search the official customs ruling databases (CROSS) for related cases. If a related case is found (very rare during the annotation of HSCODECOMP), the corresponding HSCode is then validated on eWTP system, followed by the minor revision. Otherwise, they refine their search queries or revisit Step 2 to adjust the extracted features (Step

---

[2]https://rulings.cbp.gov/

Figure 3. The pipeline for human experts to annotate the HSCodes, including two human experts for HSCode annotation (Step 1 to 5) and one additional expert for quality validation (Step 6).

3); (4) For products without any related cases, experts execute human-written hierarchical decision rules to apply tariff rules, and determine the appropriate HSCode (Step 4); (5) Finally, experts verify the final identified HSCodes on the eWTP website to ensure its validity (Step 5).

**Human Expert Validation.** As shown in the Figure 3 (Step 6), when the HSCodes assigned by two experts match, the code is accepted. When they disagree, a senior tariff expert reviews both annotations to determine the correct HSCode. If neither annotation is valid, the instance is excluded from the dataset. Finally, we collect 632 products with their human-annotated corresponding HSCodes, spanning 32 first-level product category defined by a large-scale ecommerce platform and 27 HS chapters defined by eWTP. Furthermore, to verify the reliability of our process, we conducted an additional quality review. A fourth senior expert, not involved in the initial annotation, re-annotated a random 10% sample of the dataset. This review shows only a 2% disagreement rate, confirming the effectiveness and consistency of our dataset construction pipeline. Please refer to Section B for more details about HSCODECOMP statistics.

## 4.2 EVALUATION METRIC

We conduct the exact match to compare the normalized HSCodes extracted from the final output against human-annotated ground truth. Our primary evaluation metric is the 10-digit HSCode accuracy, which measures whether the predicted code exactly matches the reference 10-digit code. Additionally, we also report accuracies at 2-digit, 4-digit, 6-digit and 8-digit levels to provide more comprehensive insights into the performance across different granularities.

## 5 EXPERIMENTS

### 5.1 EXPERIMENTAL SETUP

We evaluate three kinds of advanced approaches on HSCODECOMP:

**LLMs/VLMs (no tools):** We test 14 foundation models (GPT-5, Gemini-2.5-PRO, GPT-4o, Kimi-K2 (Team et al., 2025), Claude Sonnet 4, DeepSeek series (DeepSeek-AI et al., 2025), Qwen variants (Yang et al., 2025), O3-mini, Nemotron-32B (Bercovich et al., 2025)) for HSCode prediction using only internal knowledge. For VLMs (GPT-4o, GPT-5, Claude Sonnet 4, Gemini 2.5 Pro), we provide product images to assess the impact of the visual information.

**Open-source Agent Systems:** We evaluate six open-source frameworks (SmolAgents (Roucher et al., 2025), Aworld (Yu et al., 2025c), Agentorchestra (Zhang et al., 2025), OWL (Hu et al., 2025b), WebSailor (Li et al., 2025b) and Cognitive Kernel (Fang et al., 2025)) using GPT-5 as the default backbone. SmolAgent is enhanced with vision capabilities via product images, while Vision Language Models are incompatible with other agent frameworks. All frameworks utilize standardized tools including web search.

Table 1. The complete results of state-of-the-art baselines in our proposed HSCODECOMP.

| Baselines | Model Type | HSCode Prediction Accuracy | | | | |
|---|---|---|---|---|---|---|
| | | 2-digit | 4-digit | 6-digit | 8-digit | 10-digit |
| **LLM/VLM-Only** | | | | | | |
| GPT-5 | VLM | 82.12 | 70.89 | 59.97 | 41.46 | 29.27 |
| Gemini-2.5-PRO | VLM | **82.28** | 71.04 | 59.02 | 40.51 | 24.21 |
| GPT-4o | VLM | 78.01 | 64.08 | 48.10 | 29.75 | 18.51 |
| Claude Sonnet 4 | VLM | 78.80 | 64.08 | 45.25 | 22.63 | 11.23 |
| GPT-5 | LLM | 82.59 | 69.78 | 56.33 | 40.98 | 28.96 |
| Gemini-2.5-PRO | LLM | 80.54 | 69.94 | 58.54 | 40.35 | 23.42 |
| GPT-4o | LLM | 75.47 | 61.55 | 45.73 | 30.06 | 18.35 |
| Claude Sonnet 4 | LLM | 78.80 | 62.97 | 44.94 | 23.58 | 11.87 |
| Kimi-K2 | LLM | 78.01 | 62.03 | 44.15 | 24.53 | 12.18 |
| DeepSeek-R1 | LLM | 77.22 | 61.71 | 38.45 | 16.77 | 6.65 |
| DeepSeek-V3 | LLM | 77.06 | 54.43 | 32.28 | 17.25 | 6.49 |
| Qwen-MAX | LLM | 71.52 | 48.58 | 24.21 | 11.23 | 3.80 |
| Qwen3-235B-A22B | LLM | 66.93 | 49.53 | 24.53 | 6.01 | 1.74 |
| O3-mini | LLM | 77.22 | 56.17 | 24.53 | 6.65 | 1.27 |
| Qwen3-32B | LLM | 64.40 | 29.27 | 8.07 | 1.27 | 0.32 |
| QWQ-32B | LLM | 66.77 | 29.11 | 4.43 | 1.42 | 0.16 |
| Qwen2.5-72B | LLM | 20.73 | 12.34 | 3.80 | 1.42 | 0.16 |
| Nemotron-32B | LLM | 43.51 | 5.70 | 0.16 | 0.00 | 0.00 |
| **Open-source Agent System (GPT-5 Backbone)** | | | | | | |
| SmolAgents | VLM | 82.06 | **72.06** | **62.38** | **52.38** | **46.83** |
| SmolAgents | LLM | **82.28** | 70.89 | 59.81 | 49.05 | 42.72 |
| Aworld | LLM | **82.28** | 70.41 | 59.18 | 48.58 | 41.30 |
| Agentorchestra | LLM | 82.12 | 70.73 | 60.44 | 47.78 | 41.30 |
| OWL | LLM | 72.63 | 61.87 | 51.58 | 41.77 | 37.34 |
| WebSailor | LLM | 81.64 | 70.56 | 57.27 | 43.98 | 35.44 |
| Cognitive Kernel | LLM | 80.06 | 69.15 | 54.59 | 40.03 | 26.42 |

**Closed-source Agent Systems:** We assess the performance of commercial systems Manus, Gemini Deep Research, and Grok DeepSearch. As these systems do not provide public APIs, we conduct manual evaluations on 49 representative examples from the HSCODECOMP benchmark, following the evaluation protocol established in prior work (Li et al., 2025b).

All systems produce standardized outputs: a single HSCode in \boxed{...} format. More implementation details appear in Appendix F.

## 5.2 MAIN RESULTS

Table 1 summarizes the performance of state-of-the-art LLM/VLM-Only models and open-source agents on HSCODECOMP. All approaches exhibit a consistent decline in accuracy as the HSCode length increases. Notably, LLM/VLM-only baselines are much worse than agent systems due to their lack of domain-specific knowledge. The best baseline, SmolAgent (GPT-5 VLM version), achieves only 46.83% 10-digit accuracy, which remains substantially below the 95% accuracy achieved by experienced human experts. To ensure a fair comparison,

Table 2. Comparison between closed-source and open-source agents.

| Agent Systems | 10-digit |
|---|---|
| Gemini Deep Researcher | 40.81 |
| Manus | 30.61 |
| Grok DeepSearch | 26.53 |
| SmolAgents (GPT-5) | **42.86** |
| Aworld (GPT-5) | **42.86** |

we evaluate both closed-source and open-source agents on the same subset of HSCODECOMP. Table 2 shows open-source agents outperform closed-source agents. Case studies reveal that closed-source agents suffer from the premature decisions and information misprocessing problems, as detailed in Section 6.4. In summary, the significant performance gap between human experts and

the top-performing agent system underscores the challenges presented by HSCODECOMP. To better understand the factors affecting the performance, we conduct three ablation studies as below.

**Ablation Study on Hierarchical Decision Rules** The hierarchical decision rules capture how human experts apply tariff rules. To assess whether agents can effectively leverage these rules, we conduct ablation experiments for following models: GPT-5, SmolAgent (GPT-5 VLM version), Aword (GPT-5 LLM version) and WebSailor (GPT-5 LLM version). As shown in Table 3, incorporating decision rules (w/ DR) decreases accuracy for both SmolAgent and WebSailor, while Aworld achieves only marginal gains compared to the baseline without rules (w/o DR). Therefore, we remove these decision rules for agents as the default setup. These results indicate that current agent systems struggle at effectively applying human-written decision rules, thereby limits their ability to utilize hierarchical tariff rules for HSCode prediction.

Table 3. The ablation study on human-written **D**ecision **R**ules (**DR**). GPT-5 is the backbone.

| Backbone Model | Model Type | 10-digit |
|---|---|---|
| SmolAgent **w/o DR** | VLM | **46.83** |
| SmolAgent **w/ DR** | VLM | 43.83↓ |
| Aworld **w/o DR** | LLM | 41.30 |
| Aworld **w/ DR** | LLM | **42.95↑** |
| WebSailor **w/o DR** | LLM | **35.44** |
| WebSailor **w/ DR** | LLM | 35.43↓ |

**Multi-modal Information Is Helpful** Table 1 and Table 4 show that most baselines achieve consistent improvements when the product images can be accessed. Case studies in Appendix I show that understanding product images improves performance by capturing visual attributes—such as material and surface features—that are not present in the textual description but are critical for classification. These attributes align with the predefined rules, leading to performance gains.

Table 4. The ablation study of the product images in SmolAgents.

| Backbone Model | 10-digit |
|---|---|
| GPT-5 **w/o Image** | 42.72 |
| GPT-5 **w/ Image** | **46.83 ↑** |
| Gemini-2.5-Pro **w/o Image** | **34.49** |
| Gemini-2.5-Pro **w/ Image** | 34.39 ↓ |
| Claude 4 Sonnet **w/o Image** | 33.70 |
| Claude 4 Sonnet **w/ Image** | **34.65 ↑** |
| GPT-4o **w/o Image** | 22.03 |
| GPT-4o **w/ Image** | **22.31 ↑** |

**Webpage Visits Decrease Agents Performance** We augment SmolAgent (GPT-5 LLM version) with the capability to visit webpages, but this leading to 10-digit accuracy decrease from 42.72% to 42.09%. Our study reveals that a large amount of webpage content overwhelms the key information, misleading the agents, while this key information can be precisely extract by search engines in the snippets. Therefore, we remove the webpage visit tool for all open-source agents as the default setup.

## 6 ANALYSIS ON HSCODECOMP

We conduct several detailed analysis: (1) Overthinking Decrease Open-Source Agent Performance (Section 6.1); (2) Effects of Backbones in agents (Section 6.2); (3) Effectiveness of Test-time Scaling (Section 6.3); (4) Failure Modes of Closed-Source and Open-Source Agents (Section 6.4); and (5) Per-category performance analysis (Section 6.5).

### 6.1 OVERTHINKING DECREASE OPEN-SOURCE AGENT PERFORMANCE

Table 1 shows WebSailor underperforms SmolAgent despite using identical models and tools. Our case studies reveal that this occurs because WebSailor encourages excessive reasoning (**Overthink**), before tool-calling. Cases in Appendix H show that WebSailor often first conduct deep reasoning to predict the full 10-digit HSCode. The errors in reasoning significantly decreases the effectiveness of tool-calling. To prove this, we created two variants for WebSailor: (1) **No-Think**: direct tool calling without thinking; and (2) **Medium-Think**: medium-level

Table 5. Agent performance with different think depth. GPT-5 LLM version is the backbone.

| Agent Systems | Think Depth | 10-digit |
|---|---|---|
| SmolAgent (GPT-5) | No-Think | 42.72 |
| WebSailor (GPT-5) | No-Think | 40.82 |
| WebSailor (GPT-5) | Medium-Think | 37.34 |
| WebSailor (GPT-5) | Overthink | 35.44 |

reasoning depth before tool-calling. Medium-think denotes a moderate reasoning depth between No-think and Overthink. Table 5 demonstrates that reducing reasoning depth improves accuracy, with No-Think nearly matching SmolAgent. This finding suggests that the primary factor contributing to performance variances among open-source agents is the reasoning depth defined in the task prompt. For HSCode prediction, minimal reasoning with frequent tool calls outperforms extensive self-reasoning. When accurate information is available through calling tools, prioritizing tool utilization over reasoning yields better results for such complex domain-specific tasks.

## 6.2 Effect of Backbones in Open-source Agent Systems

This subsection analyze the effects of the backbone LLM in the performance of agent systems. Specifically, we evaluate the performance of SmolAgents system implemented by four different backbone LLMs: GPT-5, Gemini-2.5-pro, Claude-4-Sonnet and Qwen-MAX. As demonstrated in Table 6, different backbone LLMs yield markedly different results in the HSCode prediction task, and GPT-5 is the best backbone model. Therefore, we choose GPT-5 as the default setup for open-source agents. More results are provided in Appendix D.3.

Table 6. The ablation study on the backbone models in SmolAgent.

| Backbone Model | 10-digit |
|---|---|
| GPT-5 | **42.72** |
| Gemini-2.5-Pro | 34.49 |
| Claude 4 Sonnet | 33.70 |
| Qwen-MAX | 17.43 |

## 6.3 Test-time scaling Cannot Improve Performance Effectively

Test-time scaling (TTS) has demonstrated significant gains in complex reasoning tasks using more inference budget (Liu et al., 2025; Guo et al., 2025; Ma et al., 2025). Given these successes, we investigate whether TTS can enhance performance on HSCODECOMP. Specifically, we evaluate two established TTS strategies (Liu et al., 2025): (1) **Majority Voting:** We implement majority voting across $K = \{1, 2, 4, 8, 16\}$ independent trials (Voting@$K$), using SmolAgent (GPT-5). Figure 4 showns that increasing $K$ yields negligible performance improvement; (2) **Self-Reflection:** We integrate a self-reflection mechanism into SmolAgent (GPT-5), enabling the model to proac-

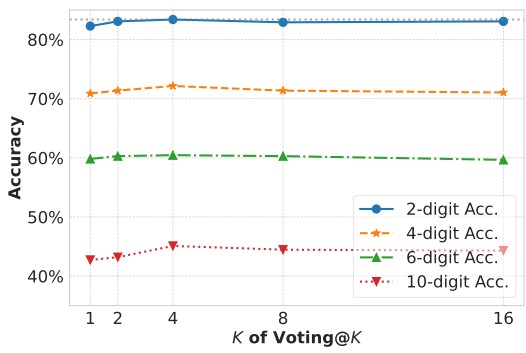

Figure 4. Majority voting experiments.

tively evaluate and revise its reasoning and actions. However, this approach slightly decreases performance from 42.72% to 42.57%. These results demonstrate a key limitation of standard TTS methods when applied to HSCode prediction, highlighting the need of more effective test-time scaling strategy for agents in hierarchical rule application.

## 6.4 Failure Modes of Closed-Source and Open-Source Agents

We perform the qualitative and quantitative analysis for closed-source and open-source agents.

**Qualitative analysis.** We identify six critical failure modes of open-source and closed-source agent systems in HSCODECOMP: (1) **Premature Decisions**: Agents commit to incorrect classification paths without collecting sufficient evidence (Table 13, Grok DeepSearch); (2) **Information Misprocessing**: Agents overlook or misinterpret key product details, indicating challenges with long-context processing (Table 12 and Figure 16); (3) **Unnecessary Self-Correction**: Agents sometimes predict correct HSCodes initially but revise them incorrectly through excessive critique (Table 12, Gemini Deep Research); (4) **Reasoning Hallucination**: Agents generate plausible but factually incorrect reasoning steps (Table 12, Grok DeepSearch); (5) **Wrong Rule Application**: Models frequently miss or misuse relevant tariff rules due to their ambiguous descriptions that confuse the reasoning process, resulting in incorrect classification decisions (Figure 18); and (6) **Lack of Domain Knowledge**: Models exhibit errors due to insufficient domain-specific knowledge, such

as misidentify silicone products as rubber instead of plastic (Figure 17). These limitations highlight that HSCODECOMP remains challenging for advanced closed-source and open-source systems.

**Quantitative analysis:** Figure 5 present the distribution of four coarse-grained failures across both LLM/VLM-Only (left) and agents (right)[3]: (1) **Outdated**: Incorrect HSCodes due to changes in tariff rules over time; (2) **Hallucination**: Invalid HSCodes that do not exist in the official coding system; (3) **Error but Valid**: HSCodes are valid and current, but differ from the ground-truth HSCodes; and (4) **Others**: Other errors like wrong output formats, reaching maximum window and wrong tool-calling. Our analysis reveals that agents significantly reduce hallucination, outdated and other errors through effective tool utilization, compared with LLMs. Consequently, the predominant error type for agents is "Error but Valid". Besides, Figure 9 also quantifies the improvements from GPT-5 to SmolAgent (GPT-5), demonstrating that the agent significantly reduces both outdated and hallucination errors.

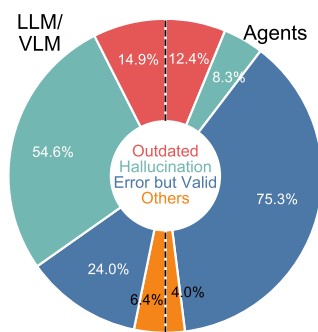

Figure 5. Failures analysis.

### 6.5 PER-CATEGORY PERFORMANCE ANALYSIS

We analyze two critical distributions across the 32 first-level product categories: (1) **Challenging Product Distribution (CID)**: the distribution of products that all baseline methods failed to correctly predict; (2) **Average Performance Distribution (APD)**: the distribution of average 10-digit accuracy across all baseline methods. Figure 6 reveals two key insights: (1) The CID indicates that the most challenging products are concentrated in long-tail categories, such as *Novelty & Special Use* (1.3%) and *Men's Clothing* (2.2%); (2) The APD shows that average accuracy across most product categories remains below 25%, with only *Hair Extensions & Wigs* achieving a relatively high accuracy of 47%. Importantly, even for most frequent categories like *Jewelry & Accessories* (13.1%), *Home & Garden* (16.8%) and *Tools* (6.2%), the average performance stays below

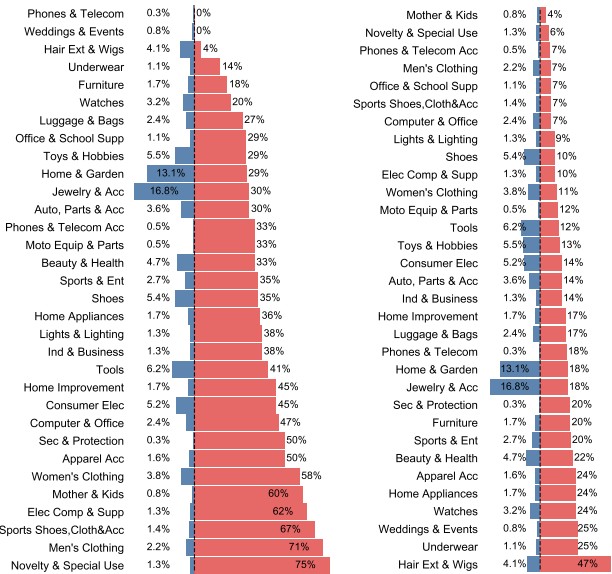

Figure 6. Both figures show the category distribution on the left blue bars. **Left:** Challenging Product Distribution (CID). **Right:** Average Performance Distribution (APD).

18%. These findings underscore the challenges presented by HSCODECOMP, highlighting the need for more robust and generalizable approaches to HSCode prediction.

## 7 CONCLUSION

We identified and addressed the critical gap in evaluating deep search agents in hierarchical rule applications. To address this gap, we introduced HSCODECOMP, the first realistic and expert-level benchmark designed to assess agents for multi-hop reasoning with hierarchical tariff rules in e-commerce domain. Our extensive evaluation revealed a substantial performance gap between current state-of-the-art agents (46.8%) and human experts (95.0%), highlighting that hierarchical rule application remains a significant challenge for existing agent architectures. We will release the HSCODECOMP to accelerate research in this crucial capability for real-world agent deployment.

---

[3]The average performance of LLM-only baselines and agent baselines are computed.

## 8 ETHICS STATEMENT

This research adheres to strict ethical guidelines regarding data privacy and fair labor. The dataset is fully anonymized and contains no personally identifiable information. The hourly wage of our human annotators is over 34.6 USD, which is much higher than average hourly wage 3.13 USD on Amazon Mechanical Turk (Hara et al., 2017). This remuneration structure was designed to provide a fair and competitive wage, acknowledging the expertise and effort required for this task and ensuring that contributors were rewarded appropriately for their work.

## 9 REPRODUCIBILITY STATEMENT

We are committed to the principles of reproducible research. Accordingly, all necessary materials, including code, benchmark dataset and other related resources will be publicly released to promote the development of the deep search agents.

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

## A    THE USE OF LARGE LANGUAGE MODELS (LLMS)

In preparing this manuscript, Qwen-MAX and ChatGPT were used solely as a writing assistant to improve grammar and clarity. The LLMs was not used for generating code, concepts, or any part of the core research methodology.

## B    DATASET DISTRIBUTION

In this section, we provide following detailed data statistics of our proposed HSCODECOMP: (1) Total unique 10-digit codes is 352; (2) Sample distribution per HSCode is shown in Table 7. Since 72.73% of HSCodes appear only once in our dataset, HSCODECOMP is not intentionally enriched for tricky or dispute-prone items; and (3) HSCode subcategory Hierarchical Coverage is shown in Table 8. Besides, Figure 7 presents the distributions of first-level product categories (left) and HSCode chapter categories (right), which closely mirror real-world product distributions. This alignment confirms that HSCODECOMP accurately reflects practical international trade scenarios, ensuring that model performance evaluations reliably generalize to real-world applications. In summary, our sampling strategy for constructing HSCODECOMP directly follows sales volume and category distributions from a major global E-Commerce platform. This approach ensures the benchmark reflects actual production distributions.

Table 7. Sample Distribution per HSCode

| Samples per Code | Count | Percentage |
|---|---|---|
| 1 | 256 | 72.73% |
| 2 | 58 | 16.48% |
| 3 | 16 | 4.55% |
| 4 | 4 | 1.14% |
| 5 | 5 | 1.42% |
| 6+ | 13 | 3.69% |

Table 8. HSCode Hierarchical Coverage

| HSCode Subcategory | Count |
|---|---|
| 2-digit (chapters) | 27 |
| 4-digit (headings) | 151 |
| 6-digit (subheadings) | 268 |
| 8-digit (country-specific) | 322 |
| 10-digit (country-specific) | 352 |

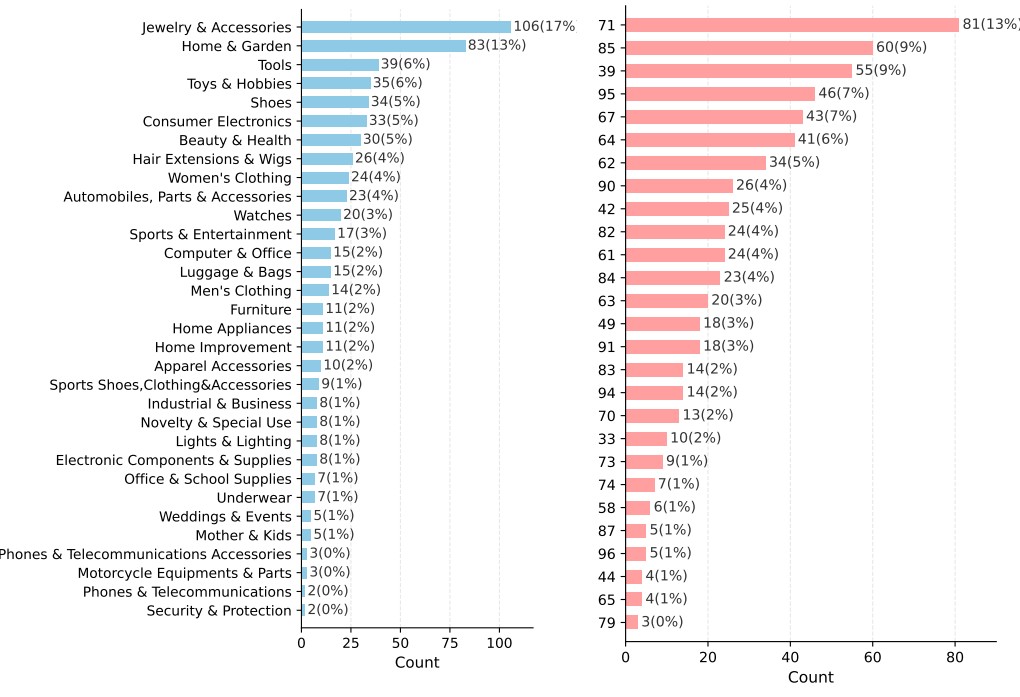

(a) First-level product categories    (b) HSCodes distribution on chapter

Figure 7. Distributions of the first-level product category and HSCode chapter categories.

## C  SEMANTIC DISTRIBUTION OF HIERARCHICAL TARIFF RULES

To assess whether the HSCode taxonomy exhibits clear semantic separation, we generate embeddings of the official English titles and notes for all HS chapters and sections using a sentence embedding model[4]. We then apply t-SNE to project these embeddings into two dimensions for visualization. As shown in Figure 8, each point represents a chapter, while each star marks a section's centroid. The visualization reveals significant semantic overlap between adjacent sections: numerous chapters appear closer to neighboring section centroids than to their own section's centroid, and section centroids themselves form overlapping clusters rather than distinct groupings. This pattern indicates that the semantic structure of hierarchical tariff rules lacks clear boundaries—adjacent sections frequently share similar vocabulary and concepts (*e.g.,* distinctions between raw materials and finished goods, or between component parts and complete articles).

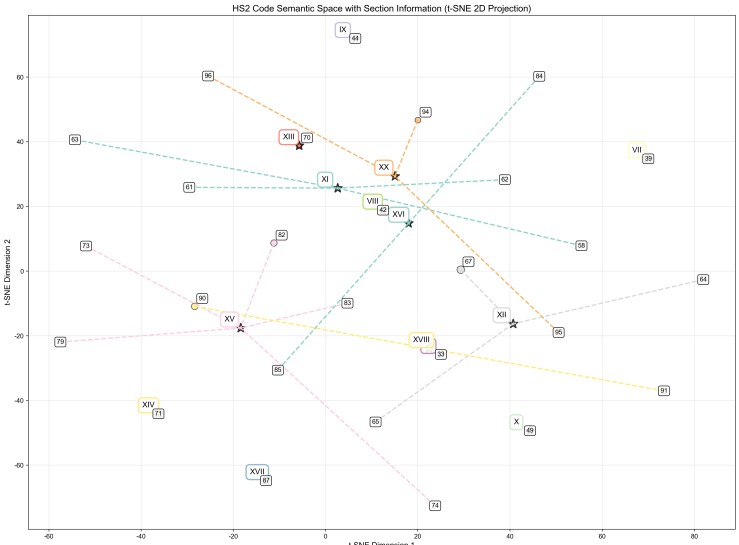

Figure 8. The semantic map of HS chapter titles and notes.

## D  ADDITIONAL EXPERIMENTAL RESULTS

### D.1  CONDITIONAL PERFORMANCE ON FINE-GRAINED SUBCATEGORIES

As shown in Table 9, we provide the conditional accuracy of partial models on Chapter, Heading, Sub-heading and Country-specific sub-categories code classification. The experimental results reveal three key insights: (1) The 2-4 digit heading level shows highest average accuracy (75.16), since its tariff rules are most clearly defined; (2) Intermediate performance drops at the 4-6 digit subheading level due to increased the number of rules and their vague descriptions in tariff rules; and (3) The steepest decline occurs at the country-specific 6-10 digit levels, where rules become highly contextual and country-specific. This hierarchical performance pattern directly supports our main finding: current agents struggle with the complex, multi-step reasoning required for precise rule application at deeper hierarchical levels. The significant accuracy drop from 64.8% (6-digit) to 35.3% (10-digit) demonstrates that correctly navigating the initial hierarchical steps does not guarantee success in the final classification, highlighting the cumulative reasoning challenges in hierarchical rule application.

### D.2  DETAILED ABLATION STUDY ON RULES

We provide the complete ablation study on used domain-specific knowledge and rules. As shown in Table 10, we remove each used rules or knowledge in WebSailor and Smolagent open-source

---

[4] https://huggingface.co/sentence-transformers/all-MiniLM-L6-v2

Table 9. The conditional performance on Chapter, Heading, Sub-heading and Country-specific codes classifiction in our proposed HSCODECOMP.

| Baselines | Model Type | HSCode Prediction Accuracy | | | |
|---|---|---|---|---|---|
| | | 2-digit Chapter Acc. | 4-digit Heading Acc. | 6-digit Sub-heading Acc. | 10-digit Country-specific Acc. |
| **LLM/VLM-Only** | | | | | |
| GPT-5 | VLM | 82.12 | 86.32 | 84.60 | 48.81 |
| Gemini-2.5-PRO | VLM | 82.28 | 86.34 | 83.08 | 41.02 |
| Claude Sonnet 4 | VLM | 78.80 | 81.32 | 70.61 | 24.82 |
| Kimi-K2 | LLM | 78.01 | 79.52 | 71.18 | 27.59 |
| DeepSeek-V3 | LLM | 77.06 | 70.63 | 59.31 | 20.11 |
| O3-mini | LLM | 77.22 | 72.74 | 43.67 | 5.18 |
| **Open-source Agent System (GPT-5 Backbone)** | | | | | |
| SmolAgents | VLM | 82.06 | 87.81 | 86.57 | 75.07 |
| Aworld | LLM | 82.28 | 85.57 | 84.05 | 69.79 |
| Agentorchestra | LLM | 82.12 | 86.13 | 85.45 | 68.33 |
| OWL | LLM | 72.63 | 85.19 | 83.37 | 72.39 |
| WebSailor | LLM | 81.64 | 86.43 | 81.16 | 61.88 |
| Cognitive Kernel | LLM | 80.06 | 86.37 | 78.94 | 48.40 |
| Average | - | 73.80 | 75.16 | 64.78 | 35.32 |

agent frameworks for solving tasks in HSCODECOMP: (1) **DR**: Human-written Decision Rules; (2) **Tariff**: hierarchical tariff rules; and (3) **CROSS** ruling database.

Experimental results reveal three critical insights: (1) Human-written decision rules impair performance: As shown in Table 3 of our paper, adding decision rules (DR) decreases accuracy by 3.00% for Smolagent (GPT-5), demonstrating current agent systems cannot effectively utilize hierarchical rule structures; (2) Tariff rules are essential and contributes most: Removing tariff rules causes a severe 10.10% drop in accuracy. Without tariff rules, agents

Table 10. Ablation study on rules and domain-specific knowledge.

| SmolAgent (GPT-5) | 10-digit Acc. |
|---|---|
| + CROSS + Tariff + DR | 43.83% |
| - DR | **46.83%** |
| - Tariff | 36.73% |
| - CROSS | 46.53% |

have to solve the task using its internal knowledge, leading to outdated and hallucination problems; and (3) CROSS provides moderate benefit: The 4.61% accuracy drop when removing CROSS indicates it offers supplementary but non-essential information. These results confirm that tariff rules are the most critical knowledge component, while human-written decision rules present the most significant challenge for current agent architectures to utilize effectively.

### D.3 MORE DETAILED EXPERIMENTS ON OPEN-SOURCE AGENTS WITH DIFFERENT BACKBONE LLMS

To investigate how the backbone LLMs affect the performance of the agent system, we conduct more detailed ablation study on four open-source agent systems, replacing the original GPT-5 backbone with Gemini 2.5 Pro. The experimental results in Table 11 indicate that GPT-5 achieves consistently better performance than the advanced Gemini 2.5 pro model on these open-source agent systems.

Table 11. The ablation study of backbone models in the open-source agent system.

| Backbone LLM | HSCode Prediction Accuracy | | | | |
|---|---|---|---|---|---|
| | 2-digit | 4-digit | 6-digit | 8-digit | 10-digit |
| SmolAgent | | | | | |
| GPT-5 | **82.28** | **70.89** | **59.81** | **49.05** | **42.72** |
| Gemini-2.5-Pro | 82.19 | 69.48 | 57.87 | 44.04 | 34.49 |
| Claude 4 Sonnet | 80.69 | 67.09 | 54.11 | 42.25 | 33.70 |
| Qwen-MAX | 77.34 | 63.23 | 42.47 | 26.62 | 17.43 |
| Aworld | | | | | |
| GPT-5 | **82.28** | **70.41** | **59.18** | **48.58** | **41.30** |
| Gemini 2.5 Pro | 79.55 | 66.97 | 54.70 | 38.79 | 29.24 |
| WebSailor | | | | | |
| GPT-5 | **81.64** | **70.56** | **57.27** | **43.98** | **35.44** |
| Gemini 2.5 Pro | 78.79 | 67.58 | 56.21 | 42.27 | 31.21 |
| AgentOrchestra | | | | | |
| GPT-5 | 82.12 | **70.73** | **60.44** | **47.78** | **41.30** |
| Gemini 2.5 Pro | **82.27** | 69.39 | 56.97 | 41.36 | 30.61 |

# E  IMPROVEMENT GAIN FROM AGENTS

This waterfall in Figure 9 chart reveals that the superior performance of SmolAgent (GPT-5) are primarily from reducing the outdated and hallucination failures, with 56 corrected samples in HSCOD-BENCH. Besides, as shown in Figure 10, it can be found that the rate of outdated and hallucination are significantly reduced in SmolAgent baseline.

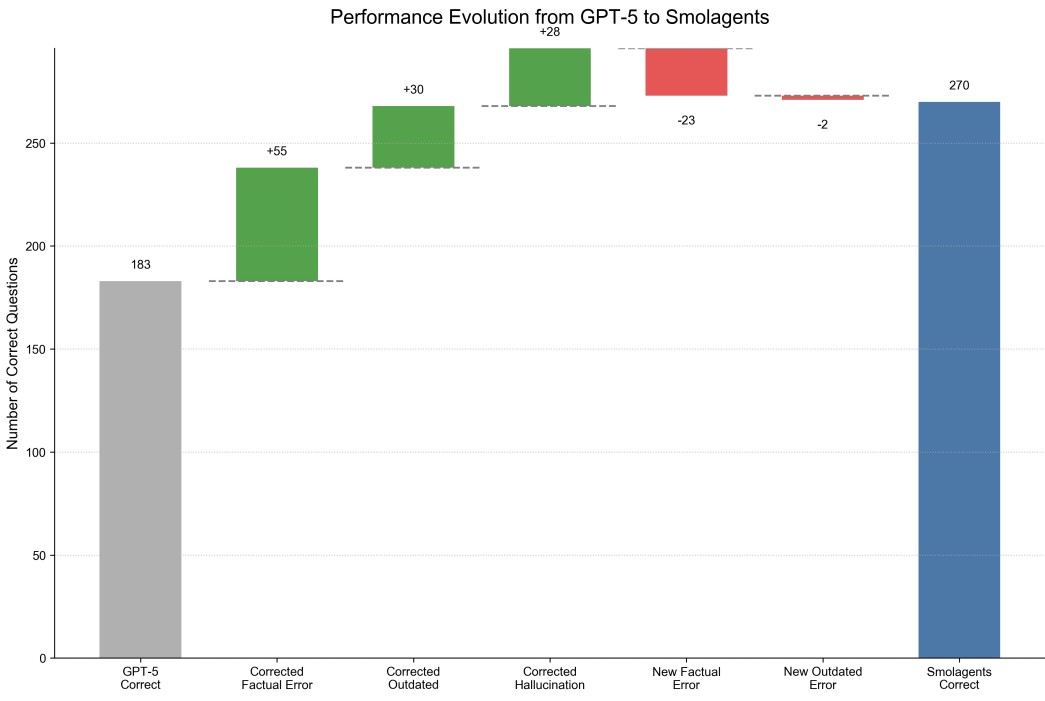

Figure 9. Details of performance gain and loss comparing GPT-5 and Smolagents.

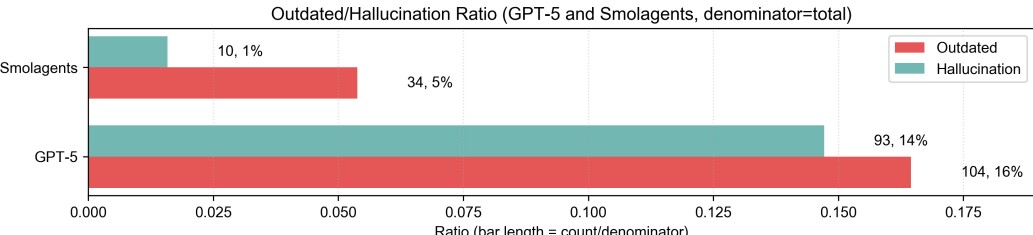

Figure 10. Outdated or hallucination ratio happended on GPT-5 and Smolagent. The numbers are number and ratio of the phenomenon.

# F    IMPLEMENTING DETAILS AND KNOWLEDGE FORMS

**Evaluation Details**: All baseline methods are equipped with search tools to access the CROSS database, hierarchical tariff rules, and other related resources, including human-written knowledge bases and hierarchical decision rules. The temperature and context window size of LLMs and agents are set to their default configurations. Moreover, as described in Section 5.2, the hierarchical decision rules, and webpage visit tool are not used during evaluation, since they do not improve the performance of open-source and closed-source agents. The multi-modal product images are used for open-source agents for SmolAgents. The hierarchical decision rules used in our prompts are illustrated in Figure F.1. The hierarchical tariff rules in the eWTP is shown in Figure 11. It can be found that the red boxes highlight implicit logical relationships in the tariff rules, such as *excluding articles of heading 8593* and *with the machines of heading 8501 or 8502*. The blue boxes highlight vague descriptions in the tariff rules, such as ... *for example* ... and ... *such as* .... These cases demonstrate that rule boundaries are ambiguous, posing significant challenges for accurate rule application by the agent. Moreover, the U.S. Customs Rulings Online Search System (CROSS) interface is shown in Figure 12. As illustrated, the CROSS website contains not only correct precedent results for product HS Codes but also numerous revoked precedents, requiring the agent to carefully evaluate information reliability. Additionally, since the precedent information is presented as plain text emails, the agent must effectively utilize contextual information to perform accurate reasoning.

**Knowledge Alignment**: All these knowledge resources in HSCODECOMP: hierarchical rules come directly from official US HTS legal notes included in eWTP, and the CROSS database contains authentic US Customs rulings. Human-written decision rules are also developed by human experts with US compliance experience.

**Timestamp-based Evaluation**: To ensure the reproducibility of our proposed HSCODECOMP dataset, all our dataset, knowledge and experimental setup are tied to the specific timestamp—June 30, 2024. This is a wide-used community standards of established benchmarks like BrowseComp Wei et al. (2025) and WideSearch Wong et al. (2025).

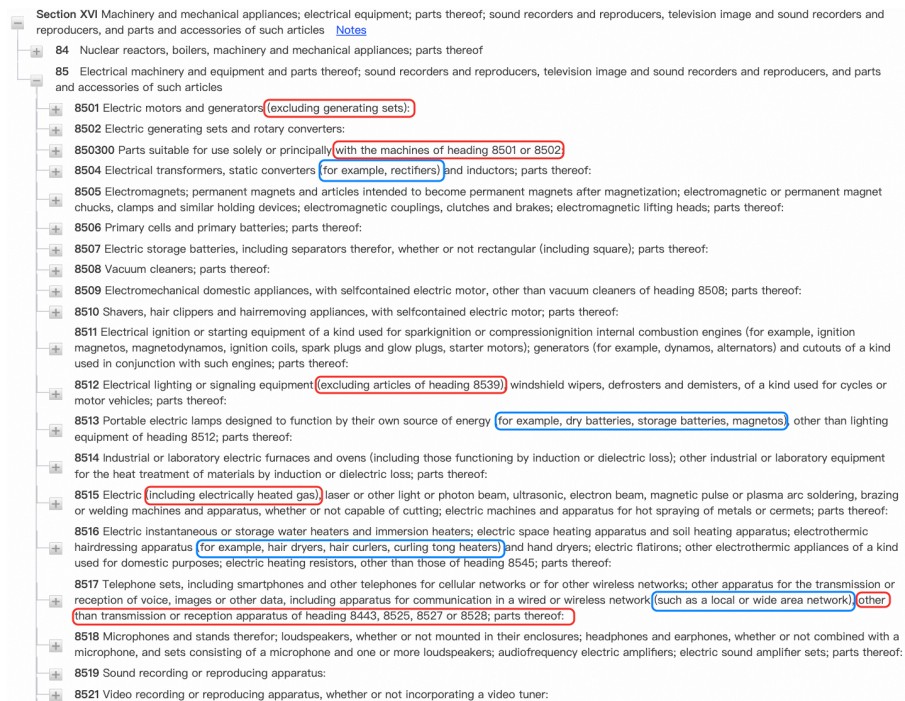

Figure 11. The case of the hierarchical tariff rules.

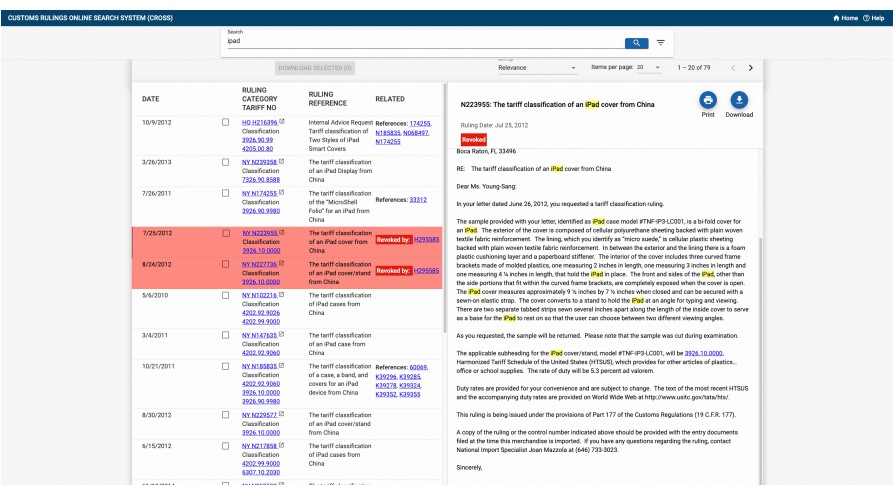

Figure 12. The case of CROSS website that contains the products rulings.

---

**Decision Rules**

**The six decision rules for hierarchical tariff rules application**
The following six rules must be applied progressively from Rule 1 to Rule 6, without skipping.

**Rule 1: Priority of Headings and Notes**
The classification of goods shall be determined primarily according to the terms of the headings (4/6-digit HS codes) and any related Notes. Subsequent rules shall only be applied if the terms of the headings and the Notes do not suffice for classification.

**Rule 2: Incomplete/Unfinished Articles and Extension to Materials/Substances**
Rule 2(a): An incomplete or unfinished article (e.g., a bicycle missing wheels), if it has the essential character of the complete article, is to be classified as the complete article.
Rule 2(b): An article consisting of a certain material or substance, which retains its original character after the addition of other materials/substances (e.g., a plastic cup with a metal base), is to be classified according to the original material.
Example: An unassembled computer motherboard (which already has the function of a motherboard) is classified under heading 8473 (parts of computers).

**Rule 3: Decision Logic for Goods Classifiable Under Multiple Headings**
When goods are classifiable under two or more headings, classification shall be effected as follows, in order of priority:
Specificity (The more specific description shall be preferred to a more general description); Essential Character (Determined by the main material, function, or use of the goods); Last in Numerical Order (If classification cannot be determined otherwise, classify under the heading which occurs last in numerical order).
Example: An electric toothbrush (which has the attributes of both a "household appliance" and an "oral hygiene tool"): Specificity: Classified as a "domestic electro-mechanical appliance" (heading 8509) rather than a "toothbrush" (heading 9603).

**Rule 4: Principle of Closest Analogy**
When goods cannot be classified by applying the preceding three rules, they shall be classified under the heading appropriate to the goods to which they are most similar.
Example: Imitation leather made from a new material (not listed in the HS) is classified as "artificial leather" (heading 3921).

**Rule 5: Packing Materials and Containers**
Rule 5(a): Packing materials/containers presented with the goods (e.g., a jewelry box), if normally sold with the goods, are classified with the goods; otherwise, they are classified separately.
Rule 5(b): Reusable packing containers (e.g., metal gas cylinders) are classified separately.
Example: A glass bottle presented with perfume is classified under the heading for perfume (3303); however, a glass bottle sold separately is classified under 7010.

**Rule 6: Hierarchical Classification at the Subheading Level**
The classification of goods in the subheadings (6-digit and subsequent HS codes) of a heading shall be determined level by level, first determining the 1-dash subheading (5-6 digits), and then successively the lower-level subheadings. At each level, classification must take into account any Subheading Notes and the relationship between subheadings at the same level.
Example: After classifying goods under heading 6205 (men's shirts), the subheading is chosen based on material (cotton, man-made fibers, etc.): 620520 (of cotton) or 620530 (of man-made fibres).

Figure 13. Decision rules defined by human experts.

# G CASE STUDY OF FAILURE MODES

We identify six critical failure modes of open-source and closed-source agent systems in HSCODE-COMP: (1) **Premature Decisions**: Agents commit to incorrect classification paths without collecting sufficient evidence (Figure 15 and Table 13-Grok DeepSearch); (2) **Information Misprocessing**: Agents overlook or misinterpret key product details, indicating challenges with long-context processing (Table 12 and Figure 16); (3) **Unnecessary Self-Correction**: Agents sometimes predict correct HSCodes initially but revise them incorrectly through excessive critique (Table 12, Gemini Deep Research); (4) **Reasoning Hallucination**: Agents generate plausible but factually incorrect reasoning steps (Table 12, Grok DeepSearch); (5) **Wrong Rule Application**: Models frequently miss or misuse relevant tariff rules due to their ambiguous descriptions that confuse the reasoning process, resulting in incorrect classification decisions (Figure 18); and (6) **Lack of Domain Knowledge**: Models exhibit errors due to insufficient domain-specific knowledge, such as misidentify silicone products as rubber instead of plastic (Figure 17). These limitations highlight that HSCODE-COMP remains challenging for advanced closed-source and open-source systems.

Moreover, we also provide one case in Figure 14, demonstrating cutting-edge agent's failure on using human-written hierarchical decision rules. This example demonstrates how explicit rules can encourage mechanical pattern-matching over substantive functional analysis—a critical failure mode in hierarchical rule application. Specifically, the rules-based approach mechanically applied Rule 1 literal description, classifying the collar under heading 8526 (radio navigational equipment) because it contains a GPS receiver. It cited ruling NY N006896 for "GPS pet locator devices", treating GPS presence as determinative of classification. However, this misapplies Rule 3 Essential Character principle. The product's essential function is not navigation but electronic pet containment and training. GPS merely serves as the technical means to enforce boundaries. The non-rules version correctly recognized this functional distinction, classifying it under heading 8543 (electrical machines with individual functions).

> **Failure of Using Human-written Decision Rules (Task#31)**

---

**Task**

**Product Title**:GPS-enabled dog fence collar that triggers alerts when pets cross virtual boundaries.

**Ground-truth HSCode**: 8526910040

**SmolAgent + GPT-5 (without human decision rules)**: Correctly predicted 8526910040, with Rule 1

**SmolAgent + GPT-5 (with human decision rules)**: Incorrectly classified as 8543709860 at the 4-digit level, with Rule 3

**Used Rule 1 and 3 can be found in Figure F.1**

**Failure Analysis:**
The rules-based approach mechanically applied Rule 1's literal description, classifying the collar under heading 8526 (radio navigational equipment) because it contains a GPS receiver. It cited ruling NY N006896 for "GPS pet locator devices", treating GPS presence as determinative of classification. However, this misapplies Rule 3's Essential Character principle. The product's essential function is not navigation but electronic pet containment and training. GPS merely serves as the technical means to enforce boundaries. The non-rules version correctly recognized this functional distinction, classifying it under heading 8543 (electrical machines with individual functions).

Figure 14. The case of failure of agent on applying hierarchical human-written rules.

**Error: Premature Decisions**

## Task

**Product Title**: PEN-F PU Leather Half Case for Olympus PEN-F Digital PENF Camera Brown/Black/Coffee

**Product Attributes**:

```
Origin:  Mainland China,
Measurement unit:  100000015,
Use:  Mirrorless System Camera,
Package size - length (cm):  15,
Material:  PU,
Model Number:  PEN-F,
Brand Name:  NiYi,
Type:  Camera Bags, Hard Bag,
each pack:  1,
Package size - width (cm):  8,
Package weight:  0.200,
Style:  handbags,
Package:  No,
Package size - height (cm):  5,
Sold in:  sell_by_piece
```

**Product Price**: 9.8 USD
**Category**: Consumer Electronics → Accessories & Parts → Camera Bags & Cases

## Search

**Search query:** HTSUS camera case 4202.92 United States 10-digit code camera cases outer surface of plastic sheeting.

## Analysis

Description of the actual code:
39 Plastics and articles thereof
3926 Other articles of plastics and articles of other materials of headings 3901 to 3914:
392690 Other:
39269099 Other
3926909989 Other
The accurate code is 3926909989. This first search query leads to a wrong direction since it decides the item is under 4202 and ignored it is PU leather which is plastic, and later turns did not realize the ignorance.

Figure 15. Early wrong search query leads to wrong result.

**Error: Real-world noise**

**Product Title**: 10/30/60PCS INS Blue Color PVC Sticker Aesthetic Hand Accounting DIY Decoration Scrapbooking Korean Stationery Supplies

**Product Attributes**:

```
Origin :  Mainland China
Shape :  malformed
Size :  M
Package size - length (cm) :  20
Material :  Plastic
Model Number :  sticky PVC stickers
LOT_NUM : 1
Hign-concerned Chemical :  None
PRODUCT_UNIT : 100000015
Brand Name :  NoEnName_Null
PACKAGE_TYPE : sell_by_piece
Package size - width (cm) :  15
Package weight :  0.030
Package size - height (cm) :  3
Age :  >3 years old
```

**Product Price**: 0.77 USD

**Category**: Office & School Supplies → Stationery Sticker → Stationery Sticker

## Search query and Prediction

**Search query:** site:rulings.cbp.gov PVC stickers classification 4911.91

**Prediction:** | 4911.91.4040 |

Classification path - Chapter 49: Printed books, newspapers, pictures and other products of the printing industry - Heading 4911: Other printed matter, including printed pictures and photographs - Subheading 4911.91: Pictures, designs and photographs - 8-digit: 4911.91.40: Other (i.e., not lithographs, not posters, and not printed over 20 years prior to import) - 10-digit: 4911.91.4040

Decision rationale - The product consists of decorative, self-adhesive PVC stickers bearing printed designs (for scrapbooking/decoration). The essential character is the printed picture/design, not the plastic material or adhesive.
- CBP consistently classifies decorative stickers (including PVC, puffy/3D, epoxy, etc.) as printed pictures under 4911.91 when the printing is the primary purpose, rather than as plastic self-adhesive plates/labels of heading 3919 (which typically applies to pressure-sensitive labels where the printing is merely incidental to labeling).
- Representative rulings:
- NY N300171 (Assorted Stickers from China): classified under 4911.91.4040 as other printed pictures, designs and photographs.
- NY N262652 (3D PVC cat stickers): classified under 4911.91.4040.
- NY N273080 (craft stickers, including PVC): classified under 4911.91.4040.
- NY N301146 (sticker set with PVC printed sheeting): printed picture/design classified in 4911.91.4040.
- The goods are contemporary decorative stickers (not printed over 20 years prior to import), so 4911.91.10.00 does not apply. They are not posters (4911.91.30) and not lithographs on paper (4911.91.20). Therefore, they fall in "Other," 4911.91.40, statistical suffix .4040.
Sources consulted
- HTSUS 4911.91 structure (USITC search page: 4911.91) and Chapter 49 PDF (2024)
- CBP Rulings: N300171, N262652, N273080, N301146 (all accessible via rulings.cbp.gov)",

## HSCode Description

**Predicted:**
49 Printed books, newspapers, pictures and other products of the printing industry; manuscripts, typescripts and plans
4911 Other printed matter, including printed pictures and photographs:
491191 Pictures, designs and photographs:
49119140 Other
4911914040 Other
**True:**
39 Plastics and articles thereof
3919 Selfadhesive plates, sheets, film, foil, tape, strip and other flat shapes, of plastics, whether or not in rolls:
391910 In rolls of a width not exceeding 20 cm:
39191020 Other
3919102055 Other

> **Analysis**
>
> The accurate code is 3919.10.2055. The material is PVC plastic, and its essential character is defined by its physical form as a self-adhesive sheet in a roll, not the incidental printing on its surface. Therefore, it cannot be classified under Chapter 49 (Printed Matter). The misclassification error stems from **real-world noise** in the product title. This noise takes the form of subjective marketing terms (e.g., **Aesthetic**, **INS**, **Korean Stationery**) and application-focused words (e.g., **Decoration**, **Scrapbooking**). This noise creates a misleading narrative that the printed design is the product's essential character. It directly influences the outcome by shifting focus from the product's objective physical nature (a plastic article) to its subjective use (a decorative image), leading the classifier down the incorrect path of Chapter 49. Instead, as an article of plastic under Chapter 39, the hierarchical rules for self-adhesive plastics in rolls point directly to 3919.10.2055.

Figure 16. Real-world noise forms and analysis.

**Error: Lack domain knowledge**

---

**Task**

**Product Title**: 20/50/100PCS Pink Nitrile Gloves Disposable Latex Powder Free Gloves for Household Cleaning Beauty Salon Working Kitchen Gloves

**Product Attributes**:

```
Origin :  Mainland China
Brand Name :  INMAICH
Type :  Disposable
each pack :  1
Measurement unit :  100000015
Use :  Cleaning
Classification :  Oversleeve
Style :  Brief
Material :  PVC
Model Number :  Nitrile Disposable Gloves
Sold in :  sellbypiece
```

**Product Price**: 11.56 USD
**Category**: Home & Garden → Household Merchandises → Disposable Household Cleaning Products

---

**Search query and Prediction**

**Search query:** HTSUS nitrile disposable gloves HS code 10-digit 4015.19 CBP ruling

**Prediction:** 4015.19.1010

Classification path - Section VII: Plastics and Articles Thereof; Rubber and Articles Thereof - Chapter 40: Rubber and articles thereof - Heading 4015: Articles of apparel and clothing accessories (including gloves), for all purposes, of vulcanized rubber other than hard rubber - Subheading 4015.19: Gloves: Other (than surgical) - Statistical suffix 4015.19.1010: Seamless, disposable gloves, of vulcanized rubber other than hard rubber

Decision rationale - Material and nature: The product is described and marketed as "nitrile gloves." Nitrile is a synthetic rubber (nitrile-butadiene rubber), which places the goods within Chapter 40 (rubber), not plastics. - Article type: Gloves for household cleaning/beauty salon/working, i.e., apparel/clothing accessories of rubber → Heading 4015. - Use: Non-surgical/industrial/household use → not classifiable as surgical gloves (excludes 4015.11). - Construction and disposability: Typical disposable nitrile gloves are dip-molded and seamless; they are explicitly described as "Disposable." Accordingly, within 4015.19, they fall under the statistical provision for seamless, disposable gloves → 4015.19.1010.

Supporting references - CBP NY H86590: classifies disposable nitrile gloves as 4015.19.1010 (seamless, disposable rubber gloves; non-surgical). - CBP NY D83273: disposable nitrile gloves classified under 4015.19.1010.

Notes - If the goods were actually made of PVC/vinyl (plastic) rather than nitrile rubber, they would instead fall in Chapter 39 (e.g., 3926.20.xx for plastic gloves). However, based on the stated product title/model ("Nitrile Disposable Gloves"), the correct US HTSUS 10-digit code is 4015.19.1010.

**HSCode Description**

**Predicted:**
40 Rubber and articles thereof
4015 Articles of apparel and clothing accessories (including gloves, mittens and mitts), for all purposes, of vulcanized rubber other than hard rubber:
- Gloves, mittens and mitts:
401519 Other:
40151911 Seamless
4015191110 Disposable
4015191150 Other
4015195100 Other
**True:**

39 Plastics and articles thereof
3926 Other articles of plastics and articles of other materials of headings 3901 to 3914:
392620 Articles of apparel and clothing accessories (including gloves, mittens and mitts):
39262010 Seamless
3926201020 Disposable

**Analysis**

This failure is a classic example of **lacking critical domain knowledge**, demonstrated by an inability to resolve **contradictory product data** and the use of an **outdated HTS code**. The analysis was fundamentally flawed because it prioritized the marketing-oriented *Product Title (Nitrile Gloves)* over the explicit technical attribute, *Material: PVC*. This **critical error** led the analyst to misclassify the item as a **rubber** product under **Chapter 40**. A domain expert would recognize that a **specific material attribute is a more reliable basis** for classification than a potentially inaccurate title, correctly identifying the item as a **plastic** article of **Chapter 39**. Compounding this mistake, the predicted code, 4015.19.1010, is an **obsolete tariff number**, a definitive sign that the analyst's **domain knowledge is not current** with the official HTSUS schedule.

Figure 17. Lack of domain knowledge

**Error: Wrong rule application**

## Task

**Product Title**: Fashion TYPE 2 DIABETIC Bracelet Soft Silicone Bracelet With Type 2 Diabetes Medical Warning Bracelet Hot Sale

**Product Attributes**:

```
Origin :  Mainland China
Shape pattern :  Round
Measurement unit :  100000015
Package size - length (cm) :  15
Fine or Fashion :  Fashion
Material :  Silicone
Gender :  Unisex
CN : Fujian
Item Type :  BRACELETS
Model Number :  SL--268
Metals Type :  None
Hign-concerned Chemical :  None
Brand Name :  NPKDS
Function :  None
each pack :  1
Package size - width (cm) :  15
Package weight :  0.013
Style :  TRENDY
Setting Type :  None
Clasp Type :  None
Compatibility :  All Compatible
Package size - height (cm) :  2
Sold in :  sell_by_piece
Bracelets Type :  Hologram Bracelets
Chain Type :  Rope Chain
```

**Product Price**: 1.71 USD

**Category**: Jewelry & Accessories → Fashion Jewelry → Bracelets

## Search query and Prediction

**Search query:** US HTS 7117.90 silicone bracelet imitation jewelry HS code medical alert bracelet
**Prediction:**
boxed7117.90.7500
Classification path - Section XIV: Natural or cultured pearls; precious or semiprecious stones; precious metals; imitation jewelry; coin - Chapter 71: Natural or cultured pearls, precious or semiprecious stones, precious metals, etc. - Heading 7117: Imitation jewelry - Subheading 7117.90: Other - Statistical breakout 7117.90.7500: Valued over 20 cents per dozen pieces or parts; other; of plastics
Decision rationale - Product is a bracelet made of silicone (a plastic), marketed as a fashion/medical warning bracelet. It is not of precious metal or stone and functions as personal adornment/identification, fitting the terms of "imitation jewelry" of heading 7117. - Material: Silicone → aligns with "of plastics." - Value: Listed price is 1.71 USD per piece, which equals 20.52 USD per dozen, clearly "valued over 20 cents per dozen." - CBP precedent supports this classification for silicone or medical ID bracelets when not of precious metal: - NY N127618 (silicone wristband): 7117.90.7500 (valued over $0.20/dozen; of plastics). - H015873 (Medical ID bracelet): classifiable under 7117.90.75 when valued over $0.20/dozen. - NY N121391 (silicone wristband): if valued not over $0.20/dozen → 7117.90.5500; otherwise use 7117.90.7500. Given this product's price, 7117.90.7500 applies.
Conclusion - The correct US 10-digit HTS code for this silicone "Type 2 Diabetes" fashion/medical warning bracelet is 7117.90.7500.",

## HSCode Description

**Predicted:**
71 Natural or cultured pearls, precious or semiprecious stones, precious metals, metals clad with precious metal, and articles thereof; imitation jewellery; coin
7117 Imitation jewelry:
711790 Other:
7117907500 Of plastics
**True:**
39 Plastics and articles thereof
3926 Other articles of plastics and articles of other materials of headings 3901 to 3914:
392690 Other:
39269099 Other
3926909989 Other

## Analysis

This failure is a classic case of **Wrong Rule Application**, where a basic rule was incorrectly prioritized over a decisive, higher-level one. The agent mistakenly applied the simple rule of **classification by apparent form**, seeing a *bracelet* and defaulting to the seemingly specific **Chapter 71 (Imitation Jewelry)**. However, the correct, overriding rule that should have been applied is the Rule of **Essential Character**, which embodies the important principle of **Function Over Form** in customs classification. In this case, the text **"TYPE 2 DIABETIC"** was the key that defined its core **function**, fundamentally changing the product's essence from an ornament to a **utilitarian medical alert device**. An article with such an overriding utilitarian function is legally excluded from the scope of **Chapter 71**. Therefore, the correct rule application mandates that the item be classified by its **function** and material under **Chapter 39**, not by its misleading physical form.

Figure 18. Wrong Rule Application.

---

### Case 1 of Closed-source Agents

**Product Title**: Forklift Safety Cage, 36"x36" inch Heavy Duty Collapsible Forklift Work Platform,1200LBS Capacity with 4 Universal Wheels

**Product Attributes**:

```
Origin:US(Origin)
Brand Name:NoEnNameNull
Installation Method:Assembly
each pack:1
Package size - width (cm):21
Measurement unit:100000015
Package size - length (cm):25
Package weight:0.900
Package size - height (cm):23
Sold in:sell_by_piece
Hign-concerned Chemical:None
```

**Product Price**: 1765.15 CNY
**Category**: Furniture → Outdoor Furniture → Garden Furniture Sets

---

Table 12. Comparative Analysis of AI Model Classifications for a Forklift Safety Cage mentioned above.

| Analysis Dimension | SmolAgent | Gemini DeepResearch | Grok DeepSearch | Manus |
|---|---|---|---|---|
| **Final HTSUS Code** | 8431.20.0000 (Correct) | 7326.90.8688 (Incorrect) | 8428.90.0290 (Incorrect) | 8427.90.0020 (Incorrect) |
| **Core Logic Explained** | Based on the core principle of HTSUS Section XVI, Note 2, the cage is a **part** as it is 'solely or principally for use with' a forklift (heading 8427). Its design, function, and identity are entirely dependent on the forklift. | The argument is based on the **'part vs. accessory' distinction**. It posits the cage is not an **'indispensable' part**, but an optional 'accessory'. Since accessories are precluded from 8431, classification defaults to its constituent material (steel). | Characterizes the cage as a **functional piece of machinery**. The rationale is that it enables a new function and **incorrectly compares it to complex attachments** with their own mechanics (e.g., rotators, clamps). | Characterizes the cage as a **complete 'aerial work platform'**. The core argument is that its **'4 universal wheels' constitute a 'mobile base'** per the legal notes, thus assembling it into a complete vehicle. |
| **Key Flaw Analysis** | This approach correctly identifies the product's primary use and applies the controlling legal note directly, which is the standard and most reliable method for classification. | This is overthinking because the model became **fixated on a complex, secondary legal nuance** (part vs. accessory) while **ignoring the more direct, primary rule** ('solely or principally for use with'), leading to an unnecessarily complicated and incorrect conclusion. | This is an analysis hallucination because the model **invents characteristics the product lacks**, effectively **treating a passive structure as an active machine**. The entire analysis is built on this fabricated, non-existent product feature. | The model misses key product information by **misunderstanding the function of a key feature** (the wheels). It correctly identifies the wheels but misses their trivial context (ground convenience), instead **mistaking them for a vehicle's chassis**, which invalidates the entire classification. |
| **Failure Modes** | **Correct** | **Unnecessary Self-Correction** | **Reasoning Hallucination** | **Information Misprocessing** |

**Case 2 of Closed-source Agents (Cylinder Shock Absorber)**

**Product Title**: ZJSHUYI HR60/80 Pneumatic Cylinder Shock Absorber Adjustable Industrial Used for Manufacturing Equipment

**Product Attributes**:

```
Origin:Mainland China
product name:Hydraulic buffer
Measurement unit:100000015
Package size - length (cm):35
working environment:Suitable for automated machinery
Hign-concerned Chemical:None
Certification:CE
Brand Name:NoEnName_Null
product material:iron
each pack:1
Package size - width (cm):8
Working Temperature:-10°C +80°C
Package weight:9.000
Features:Reduce vibration and noise, increase output and extend
machine life
Package size - height (cm):5
Sold in:sell_by_piece
Operating temperature:-10°C +80°C
```

**Product Price**: 16.73 USD
**Category**: Industrial & Business → Industrial Spare Parts → Industrial Hardware

Table 13. Comparative Analysis of AI Model Classifications for a Cylinder Shock Absorber mentioned above.

| Analysis Dimension | SmolAgent | Gemini DeepResearch | Grok DeepSearch | Manus |
|---|---|---|---|---|
| **Final HTSUS Code** | 8487.90.0080 (Correct) | 8412.21.0075 (Incorrect) | 8302.49.6085 (Incorrect) | 8412.31.0080 (Incorrect) |
| **Core Logic Explained** | Based on the hierarchical structure of HTSUS Chapter 84, the shock absorber is a **generic machinery part**. After systematically eliminating more specific headings, it correctly classifies the item in the **residual heading 84.87** for parts "not elsewhere specified." | Characterizes the product as an **active hydraulic motor**. The logic is that because it is a linear-acting hydraulic device, it must be a "motor" under heading 84.12, which is an apparatus that **generates force or motion**. | It correctly identifies the passive function but then classifies it as a **simple base metal fitting**. The argument is that since it's not a motor, its classification defaults to a general heading for **common hardware and accessories**. | Characterizes the product as an **active pneumatic motor**. The rationale is based on the keyword "Pneumatic Cylinder" in the title, concluding it must be an **actuator that performs work** under heading 84.12. |
| **Key Flaw Analysis** | This approach correctly identifies the product's non-specific nature and applies the HTSUS's hierarchical structure and residual headings, which is the standard and most reliable method for such goods. | This is a fundamental misunderstanding of the product's function, as the model **mistakes a passive energy-dissipating device (a damper) for an active power-generating device (a motor)**. It confuses braking with accelerating. | The model makes a decision with insufficient information about HTSUS structure because it **fails to consider the critical distinction between Chapter 83 (simple fittings) and Chapter 84 (machinery)** and thus **underestimates the product's nature** as a piece of machinery. | This is a fundamental misunderstanding of the product's function, as the model is **misled by an inaccurate keyword** and **mistakes a passive damper for an active motor**, ignoring contradictory product attributes. |
| **Failure Modes** | **Correct** | **Information Misprocessing** | **Premature Decisions** | **Information Misprocessing** |

## H  CASE STUDY OF OVERTHINKING

**Case 1: WebSailor Overthink.**

---

**Task**

**Product Title**: Men's and Women's Fashion Printed T-shirts, Men's and Women's Fashion Shirts, WWE John Scene, Winning This Day , Summer, New

**Product Attributes**:

```
Origin :  Mainland China
Package size - length (cm) :  10
Gender :  Men
Fabric Type :  Broadcloth
Hign-concerned Chemical :  None
Special Features :  Breathable,Lightweight
Applicable Scene :  Daily
Place Of Origin :  China (mainland)
each pack :  1
Applicable Season :  Spring And Summer
Style :  Casual
Sleeve Length(cm) :  short (4-16inch)
Package size - height (cm) :  10
Collar :  O-Neck
Measurement unit :  100000015
Tops Type :  TEES
Material :  POLYESTER
Item Type :  tops
Craft of Weaving :  Knit
CN : Jiangxi
Brand Name :  NoEnNameNull
Package size - width (cm) :  10
Product Care Instructions :  Machine Wash
Hooded :  No
Pattern Type :  3D
Package weight :  0.200
Closure Type :  Pull On
Sold in :  sellbypiece
```

**Product Price**: 74.96 CNY
**Category**: Men's Clothing → Tops & Tees → T-Shirts

## First step

**Websailor:**
*We need the precise 10-digit HTSUS classification for a men's knit T-shirt of man-made fiber (polyester). Heading 6109 covers T-shirts, singlets and other vests, knitted or crocheted. Subheading splits by material: 6109.10 (cotton) vs 6109.90 (other fibers). Our item is polyester, so 6109.90. Within 6109.90, the US splits by fiber (man-made fibers vs other), and by sex (men's/boys vs women's/girls), often also by "underwear" vs "other." The product is an outerwear short-sleeve printed T-shirt (not underwear), for men. The common 10-digit code used by importers and CBP rulings for men's/boys' T-shirts of man-made fibers, other than underwear, is **6109.90.1050 (men's or boys')**. I will confirm using CBP rulings and classification references via web search.*
name : search , arguments : query : [ HTSUS **6109.90.1050 men t-shirt** man-made fibers , rulings.cbp.gov 6109.90.1050 , CBP ruling men polyester knit T-shirt HTSUS , HTS 6109.90 man-made fiber T-shirts men boys underwear other , HTSUS code men polyester T-shirt 10-digit ]
**Smolagents:** search query:
'HTSUS 6109.90 T-shirts of man-made fibers 10-digit statistical suffix men

## HSCode Description

**Websailor wrong prediction:**
61 Articles of apparel and clothing accessories, knitted or crocheted 6109 Tshirts, singlets, tank tops and similar garments, knitted or crocheted: 610990 Of other textile materials: 61099010 Of manmade fibers 6109901050 Women's (639)

**Smolagents correct prediction:**
Section XI Textiles and textile articles Notes 61 Articles of apparel and clothing accessories, knitted or crocheted 6109 Tshirts, singlets, tank tops and similar garments, knitted or crocheted: 610990 Of other textile materials: 61099010 Of manmade fibers 6109901007 Men's (638)

**Case 2: WebSailor Overthink and directly answer.**

---

**Task**

**Product Title**: SPRAY GUN with adapter 400cc Mix tank 1.3/1.8mm nozzle air spray gun with quick-connect wash-free tank paint Mixing Cup sprayer

**Product Attributes**:

```
Origin :  Mainland China
Power Type :  Pneumatic
Measurement unit :  100000015
Battery Included :  No
Use :   Commercial Manufacture
Cup Volume :  400cc
Package size – length (cm) :  30
Nozzle Diameter :  1.3mm
Model Number :  GTI
Hign-concerned Chemical :  None
Feed Type :  Gravity
Certification :  CE
Brand Name :  NTOOLS
Type :  HVLP
each pack :  1
Package size – width (cm) :  22
Package weight :  1.300
No-Load Speed :  1200rpm
Application :  Paint Spray Gun
Package size – height (cm) :  20
Sold in :  sellbypiece
```

**Product Price**: 98.0 USD
**Category**: Tools → Power Tools → Spray Gun

---

**First step**

**Websailor:**
*We need the precise 10-digit US HS (HTSUS) code for a pneumatic paint spray gun with a cup (HVLP gravity-feed). These are generally classifiable under Chapter 84, heading 8424, which covers mechanical appliances for projecting or spraying liquids or powders. Specifically, subheading 8424.20 covers "Spray guns and similar appliances." The 10-digit breakouts in the HTSUS usually distinguish paint spray guns from other spray guns (e.g., agricultural, steam/sand blasting, etc.). Based on industry practice and CBP rulings, paint spray guns (including HVLP air spray guns with attached cups) fall in 8424.20.10 at the 8-digit level, with a **final** statistical suffix commonly .90 for "Other" paint spray guns (not parts). Therefore, the full 10-digit code is **8424.20.1000** or **8424.20.1090** depending on the structure. Recalling the U.S. HTSUS, the statistical suffix for "Spray guns and similar appliances: Spray guns: Paint spray guns" is **8424.20.1000** (often the sole 10-digit). To ensure accuracy, we align with typical import classifications for HVLP automotive paint spray guns with gravity cups, which are consistently entered under **8424.20.1000**. No electrical drive; compressed air powered; used for paint spraying—squarely within that provision. No need to use other headings like 8467 (tools) because 8424 is specific to spraying devices. Now present classification path and rationale and the final 10-digit code in LaTeX box.* 8424.20.1000 *... **SmolAgents:***
search query: HTS 842420 paint spray guns 10 digit statistical suffix

> **HSCode Description**
>
> **Websailor wrong prediction:**
> 84 Nuclear reactors, boilers, machinery and mechanical appliances; parts thereof 8424 Mechanical appliances (whether or not hand operated) for projecting, dispersing or spraying liquids or powders; fire extinguishers, whether or not charged; spray guns and similar appliances; steam or sand blasting machines and similar jet projecting machines; parts thereof: 842420 Spray guns and similar appliances: 8424201000 Simple piston pump sprays and powder bellows
> **Smolagents correct prediction:**
> 84 Nuclear reactors, boilers, machinery and mechanical appliances; parts thereof 8424 Mechanical appliances (whether or not hand operated) for projecting, dispersing or spraying liquids or powders; fire extinguishers, whether or not charged; spray guns and similar appliances; steam or sand blasting machines and similar jet projecting machines; parts thereof: 842420 Spray guns and similar appliances: 8424209000 Other

## I  CASE STUDY OF MULTI-MODAL INFORMATION IN AGENT SYSTEMS

This section presents experimental case studies comparing SmolAgent (GPT-5) performance with and without product image processing.

Table 14. Vision vs Non-vision: Four Representative Cases with Key Evidence and Reasons

| Image | Product Information | Non-vision HS (10-digit) + brief rationale | With-vision HS (10-digit) + brief rationale | Vision advantage (reason category) |
|---|---|---|---|---|
|  | **Title:** Royal Wedding Bouquet Rhinestone Bride and Bridesmaid Hand Flowers Hand-made Bridal Bouquet. **Category:** Artificial Decorations. **Attributes:** Material: **Silk (natural textile)**, not plastics; bouquet of artificial flowers with rhinestones. | **6702.90.3500** (Artificial flowers of *man-made fibers*). Assumes "silk flowers" = polyester; treats bouquet as MMF artificial flowers. | **6702.90.6500** (Artificial flowers, *other than man-made fibers*). Images/text indicate **silk fabric (natural fiber)**; rhinestones do not change essential character (GRI 3(b)). | **Material/fiber identification** (natural silk vs assumed polyester). |
|  | **Title:** OTF knife parts, aviation aluminum tactical handle kit. **Category:** Hand Tools / Knife accessories. **Attributes: Handle/parts only**, no blade present; aluminum body with clip, actuator, screws. | **8211.93.0035** (Folding/pocket knives). Interprets listing as a *complete folding knife* rather than components. | **8211.95.9000** (Handles of base metal, other). Visuals confirm **no blade**; essential character is the base-metal handle assembly (parts provision applies). | **Completeness/parts identification** (parts-only vs whole article). |
|  | **Title:** Acrylic buckle, beaded/openwork shoulder bag with inner pouch. **Category:** Women's Handbags. **Attributes:** Outer surface is **beads/openwork lattice (ABS/acrylic)**, not plastic sheeting; linen pouch is interior. | **4202.22.1500** (Handbags with outer surface of *sheeting of plastics*). Assumes exterior is plastic sheeting based on "ABS." | **4202.29.9000** (Other). Images show **beads/rings openwork**, not "sheeting of plastics"; classification by outer surface (Additional U.S. Note 2 to Ch. 42). | **Structural outer-surface feature** (beads/openwork vs plastic sheeting). |
|  | **Title:** The Simpsons character collectible paper cards. **Category:** Printed matter / collectibles. **Attributes:** Cards bear **pictures only**; no rules-based deck specified; **no lithographic process evidence**. | **4911.99.6000** (Other printed matter, often linked to lithographic printing). Assumes lithographic process without explicit evidence. | **4911.91.4040** (Pictures, designs and photographs; other). Visuals confirm **picture-only cards** and lack of lithographic evidence/criteria; not a playing-card game. | **Process/evidence-based exclusion** (no lithography evidence; pictures-only). |