# OpenReview forum: "HSCodeComp: A Realistic and Expert-level Benchmark for Deep Search Agents in Hierarchical Rule Application"
_ICLR.cc/2026/Conference — ICLR 2026 Conference Withdrawn Submission_

### Official Review · Reviewer_4L3J · 2025-10-28

**Soundness:** 2
**Presentation:** 4
**Contribution:** 2
**Rating:** 4
**Confidence:** 4

**Summary:**

This paper introduces HSCodeComp, a benchmark of 632 expert-labeled products for assigning 10-digit customs tariff codes. The authors formulate the problem as high stakes for trade compliance and demonstrate a noticeable gap for LLMs, the best model/agent is ~46.8% exact match at 10 digits versus ~95% for human experts. The paper also reports accuracy at coarser HS levels (2/4/6/8/10 digits), shows common failure modes, and argues that automated tariff classification is still far from solved. The paper argues this gap demonstrates that current LLM agents are not yet reliable for trade classification.

**Strengths:**

1. Practical task with real world impact. Tariff classification is a hierarchical categorization problem very similar to other industry classification specifications. The paper justifies the problem on how the classification directly affects duties, compliance risk, and audits.
2. Furthermore, the current gap between humans and model/agents is demonstrated: top LLM agents still fail ~50% of the time at the final 10-digit code.
3. Clear Expert-created workflow for ground truth: annotators gather product attributes (materials, intended use, etc.), consult official rulings/decision rules, resolve disagreements, and escalate difficult items. This data is not easily made via synthetic generation.
4. Layered evaluation is reported at 2/4/6/8/10 digits that’s useful for probing heretical failures in the classification.
5. Error analysis names specific failure modes (“valid but not chosen,” “outdated code,” material confusion like silicone vs. rubber) which is useful in grounding the nuances of the task in relation to alignment with language.

**Weaknesses:**

1. The paper evaluates hierarchical levels, but it’s not explicit whether HSCodeComp is meant to be a flat 10-digit prediction task, or a hierarchical decision process / constrained decoding task. Would this be simpler if the workflow commits to a 2-digit chapter, then refine constrained of the child nodes? That needs to be made explicit for reproducibility and fairness in future comparisons.
2. Metrics could be improved. Currently the paper uses exact-match, even if the model picked a code in the correct branch and only got the last two digits wrong. Furthermore, top n accuracy could be provided as an appendix. It would be useful to quantify near misses, sibling confusion, or a hierarchical distance for the analysis. The qualitative examples hint that these near misses are common. This weakens the interpretability of the 46.8% number.
3. Lack of a constrained-decoding/structured prediction  baseline. From my understanding, the baseline agents are allowed to output nonexistent or structurally invalid 10-digit codes, or codes that don’t match their own predicted parent. Would a trivial hierarchical decoder (predict chapter, restrict to valid children, backtrack if invalid) would cut out a whole class of hallucination errors? Not including that baseline makes it harder to tell how much of the gap is deep legal reasoning vs. just lack of structural constraints. Furthermore could this be a language vs token misalignment, and if the numerical codes were replaced with text labels, would the gap still persist?
4. Temporal stability is unclear, HTS codes change over time: some codes get split/retired, and new product categories appear. The paper discusses an outdated code failure mode. But it does not clearly state which HS/HTS revision date is considered authoritative for HSCodeComp, nor how future updates will be versioned. Without explicit versioning or a way to update future categorizations, long-term benchmarking and replication will be shaky. We see this in GICS classifications
5. Representativeness and coverage is not clear, the dataset spans 27 HS chapters and 32 e-commerce categories, but in the dataset, how many unique 10-digit leaves are represented, how many examples per leaf, and whether this mix reflects everyday brokerage volume vs. being intentionally enriched for tricky, dispute-prone items. This matters for how generalizable the reported agent accuracy is on this set versus actual in production distributions.
6. Rules hurting the agent needs one concrete example. The paper claims that giving agents human-written tariff “decision rules” sometimes degrades performance. That’s interesting and believable given similar nuances in legal and finance, but the paper doesn’t walk through a single case. One worked example would make that claim much more convincing.

**Questions:**

1. Is the intended task definition a flat 10-digit classification or hierarchical code selection step by step? Did the authors evaluate these two modes of classification?
2. Can you quantify “near misses”? How often is the model correct through 6 or 8 digits but off at the final branch? That seems essential for interpreting the ~46.8% 10-digit score. Can you quantify some distance metric in branches/leaves?
3. Have you tried a constrained or hierarchical decoder baseline, or a structured outputs approach that only allows valid descendants and blocks nonexistent or outdated codes? Did you experiment by replacing numerical codes with enums or hard string labels for language grounding? How much could this be a side effect of tokenization?
4. Which HTS revision (with date) are your gold labels tied to? And do you plan to release HSCodeComp as versioned snapshots over time to track tariff changes? How are you planning to update the benchmark when codes change in the future?
5. How many distinct 10-digit codes are in HSCodeComp, and what does the per-code sample count look like? Is this mostly one-shot per code, or do some leaves recur?
6. Could you include one concrete example where agent performance degraded when given a human tariff rule, and why? This is an interesting failure mode related to the task and can help future work understand where the gap stems from with regards to instruction following.

---

> ### Author Response · Authors · 2025-11-20
> **Response 1/2**
>
> Thank you for your thoughtful review and valuable feedback. First of all, we believe there are some misunderstandings in your review comments (First three weaknesses and questions) about our task formulation and evaluation that we'd like to clarify directly.
>
> ### **Weakness 1/Question 1 & Weakness 3/Question 3: Task formulation is explicitly hierarchical—not flat 10-digit prediction**
> Our benchmark explicitly evaluates hierarchical multi-hop reasoning with tariff rules, not flat 10-digit prediction.
>
> **_As stated in Section 3 (lines 85-88): "Accurately predicting the exact 10-digit HSCode presents significant challenges: agents must perform multi-hop hierarchical reasoning with complex tariff rules while processing noisy product descriptions."_**
>
> All evaluated agents follow this hierarchical decision process:
> * They search official tariff databases and doman-specific knowledge at each intermediate steps
> * They apply hierarchical rules to narrow down valid paths
> * They select one code and get into the next level/sub-category in HSCode taxonomy.
>
> Figure 5 quantifies this process: agents reduce hallucination errors from 54.6% (LLMs alone that only use its inner knowledge for prediction) to just 8.3% through proper rule application.
>
> **Furthermore, we want to emphasize that a flat 10-digit prediction baseline is practically infeasible.** For example, our prelimiary study reveal that Qwen3-8B model trained on 175K past customs rulings achieved only 0.16% accuracy due to the extreme long-tail 10-digit classification distribution.**
>
> ### **Weakness 2 & Question 2: Evaluation metrics have already included fine-grained 2/4/6/8-digit hierarchical metrics (Q2).**
> **A2**: Contrayed to your claim **_"... even if the model picked a code in the correct branch and only got the last two digits wrong ..."_**, our evaluation already includes hierarchical accuracy metrics at multiple granularities: **Section 4.2 and Table 1 report exact match accuracy at 2-digit, 4-digit, 6-digit, 8-digit, and 10-digit levels precisely to capture "near misses" you're referring to.**
>
> As for your question _"How often is the model correct through 6 or 8 digits but off at the final branch? "_, these results could easily obtain from Table 1, and we present the results in "General Response". Here are some summarized highlight results:
> the average accuracy across all baselines is: 73.8% (0-2 digit chapter), 75.2% (2-4 digit heading), 64.8% (4-6 digit subheading), and 35.3% (6-10 digit country-specific code). This pattern reveals that:
> 1. The 4-digit heading level shows highest accuracy because it represents the primary classification structure where tariff rules are most clearly defined
> 2. Performance drops at the 6-digit subheading level due to increased the number of rules and vague descriptions
> 3. The steepest decline occurs at the country-specific 8-10 digit levels, where rules become highly contextual and country-specific
>
> This hierarchical performance pattern directly supports our main finding:
> * **current agents struggle with the complex, multi-step reasoning required for precise rule application at deeper hierarchical levels.**
> * **The significant accuracy drop from 64.8% (6-digit) to 35.3% (10-digit) demonstrates that correctly navigating the initial hierarchical steps does not guarantee success in the final classification, highlighting the cumulative reasoning challenges in hierarchical rule application.**
>
> In summary, we believe there are several misunderstandings in your review comments, and we hope our response could solve your concern and help you better understand our work.
>
> ---
>
> ### **Weakness 4 & Question 4: Temporal stability is unclear, HTS codes change over time: some codes get split/retired, and new product categories appear. The paper discusses an outdated code failure mode ... / Which HTS revision (with date) are your gold labels tied to? And do you plan to release HSCodeComp as versioned snapshots over time to track tariff changes? How are you planning to update the benchmark when codes change in the future?**
> **A4:** Thank you for this insightful question. We apologize for not explicitly specifying the version/timestamp of tariff rules. In the revised version (Appendix F), we have clearly documented that all gold labels, taiff rules and CROSS rulings are tied to the specific timestamp **June, 30 2025**. This is a wide-used community standards of established benchmarks like BrowseComp [1] and WideSearch [2], and all agent queries restricted to knowledge available before **June, 30 2025**.
>
> We have added these details in our revisions and hope our response could solve your concern.
>
> > [1] BrowseComp-ZH: Benchmarking Web Browsing Ability of Large Language Models in Chinese
> >
> > [2] WideSearch: Benchmarking Agentic Broad Info-Seeking

---

> ### Author Response · Authors · 2025-11-20
> **Response 2/2**
>
> ### **Weakness 5 & Question 5:  Representativeness and coverage is not clear ...; How many distinct 10-digit codes are in HSCodeComp, and what does the per-code sample count look like? Is this mostly one-shot per code, or do some leaves recur?**
> **A5**: Thank you for raising these important points about dataset statistics. We have provided the exact statistics below:
> * Total unique 10-digit codes: 352
> * Sample distribution per code:
>
> | Samples per Code | Count | Percentage |
> |------------------|-------|------------|
> | 1                | 256   | 72.73%     |
> | 2                | 58    | 16.48%     |
> | 3                | 16    | 4.55%      |
> | 4                | 4     | 1.14%      |
> | 5                | 5     | 1.42%      |
> | 6+               | 13    | 3.69%      |
>
> Since 72.73% of HSCodes appear only once in our dataset, HSCodeComp is not intentionally enriched for tricky or dispute-prone items.
>
> * HSCode subcategory Hierarchical Coverage
>
> | HSCode Subcategory          | Count |
> |-----------------------------|------:|
> | 2-digit (chapters)          |    27 |
> | 4-digit (headings)          |   151 |
> | 6-digit (subheadings)       |   268 |
> | 8-digit (country-specific)  |   322 |
> | 10-digit (country-specific) |   352 |
>
> These statistics have been added to Section 4.1 in our revision to improve transparency. Regarding real-world representativeness: **Our sampling strategy directly follows sales volume and category distributions from a major global e-commerce platform. This approach ensures the benchmark reflects actual production distributions.** We hope these statistics could solve your concern about the "Representativeness and coverage" of our HSCodeComp benchmark, and these details have been added in our revisions (Appendix B, line 711 to 721).
>
> ---
>
> ### **Weakness 6 & Question 6: Rules hurting the agent needs one concrete example. The paper claims that giving agents human-written tariff “decision rules” sometimes degrades performance ...; Could you include one concrete example where agent performance degraded when given a human tariff rule, and why?**
> **A6**: Thank you for this constructive suggestion. In our revision, we have included a concrete example from HSCodeComp (Question #31). This case demonstrats how human-written decision rules degrade agent performance. Below are detailed analysis of this case.
>
> ---
> **Task #31: GPS-enabled dog fence collar that triggers alerts when pets cross virtual boundaries.**
> * **Ground truth: 8526910040**
> * **SmolAgent + GPT-5 (without human decision rules): Correctly predicted 8526910040, using Rule 1**
> * **SmolAgent + GPT-5 (with human decision rules): Incorrectly classified as 8543709860 at the 4-digit level, using Rule 3**
> * **Used Rules 1 and 3 can be found in Figure 13 in our paper**
>
> **Failure Analysis:** The rules-based approach mechanically applied Rule 1's literal description, classifying the collar under heading 8526 (radio navigational equipment) because it contains a GPS receiver. It cited ruling NY N006896 for "GPS pet locator devices," treating GPS presence as determinative of classification.
>
> However, this misapplies Rule 3's Essential Character principle. The product's essential function is not navigation but electronic pet containment and training. GPS merely serves as the technical means to enforce boundaries. The non-rules version correctly recognized this functional distinction, classifying it under heading 8543 (electrical machines with individual functions).
>
> This example demonstrates how explicit rules can encourage mechanical pattern-matching over substantive functional analysis—a critical failure mode in hierarchical rule application.
>
> In summary, thank you for helping strengthen this important point in our paper. We have added this case in Appendix G in our revision.

---

### Official Review · Reviewer_rzk4 · 2025-10-31

**Soundness:** 3
**Presentation:** 4
**Contribution:** 3
**Rating:** 6
**Confidence:** 3

**Summary:**

The paper releases \textit{HSCODECOMP}, a benchmark of 632 expert‑annotated e‑commerce products (27 HS chapters, 32 first‑level categories) to test agents’ ability to apply hierarchical tariff rules and predict 10‑digit HS/HTS codes from noisy, multi‑modal inputs (title, attributes, image, price, category, URL). Baselines cover 14 LLM/VLMs and six open‑source agents (plus three closed‑source systems on a 49‑item subset); the best system (SmolAgents + GPT‑5 (VLM)) attains 46.83% 10‑digit accuracy vs. 95.0% for human experts (Table 1, p.6; Fig. 1, p.2). The paper analyzes “overthinking,” test‑time scaling, image utility, and failure modes, and promises to release code and data. Claimed contributions: (i) a realistic, expert‑level benchmark for rule application (their “Level‑3” knowledge); (ii) a multi‑expert annotation/validation pipeline; (iii) broad LLM/agent baselines with analyses of think‑depth, images, and test‑time scaling

**Strengths:**

This paper tackles a timely and important challenge: applying rules for HS code classification rather than relying on open-ended retrieval. The motivation and problem space are clearly illustrated in Figure 1 (left side, page 2).
The dataset and setup are realistic—inputs combine noisy product listings, structured attributes, images, and URLs—and ablation studies show that including images improves accuracy in several scenarios (Table 4, page 7; Table 10, page 36).
The data labeling process is carefully designed: two experts annotate each item, a senior adjudicator resolves disagreements, and a 10% spot check shows only 2% disagreement (Figure 3, page 5).
The authors compare a wide range of baseline models and report results consistently at all HS code levels (2-, 4-, 6-, 8-, and 10-digit). The benchmark includes strong models such as GPT-4o and Qwen variants (Table 1, page 6).
Finally, the paper provides thoughtful analysis, including (i) a study of “overthinking” behavior versus tool use (Table 5, page 7); (ii) an investigation of why scaling model size at test time brings limited gains (Figure 4, page 8); and (iii) a useful taxonomy of common failure types that highlights cases where predictions are “error-but-valid” (Figure 5, page 9).

**Weaknesses:**

First, the current evaluation metric is too narrow. It only counts exact 10-digit matches as correct, even when the model predicts a valid but slightly different code. The authors themselves note that many predictions are “Error-but-Valid.” This shows a need for more flexible metrics—such as hierarchical distance, agreement at higher HS code levels (2, 4, 6, or 8 digits), and a rule-consistency score. As it stands, many reasonable answers are unfairly marked wrong (Section 4.2, p. 5; Figure 5, p. 9).

Second, there is a factual mistake in the HS taxonomy explanation. Section 3 (“Output,” p. 4) incorrectly claims that “the last four digits (from 6 to 10) are country-specific,” when in fact only digits 7–10 vary by country, while the 6-digit level is globally standardized. This should be corrected for accuracy.

Third, the source and authority of the “rules” used are ambiguous. The paper relies heavily on eWTP tariff rules (Figures 11–12; Section 4.1) but doesn’t explain how they align with official WCO or HTS legal notes or with U.S. CROSS rulings. Depending on a commercial taxonomy may lead to discrepancies with authoritative references.

Forth, the evaluation has fairness issues. Closed-source models were tested on only a 49-item subset (Table 2, p. 6), making their results not directly comparable to the 632-item open benchmark. The authors also disabled webpage retrieval because it slightly reduced accuracy, yet this choice makes the task less realistic for genuine research workflows.

**Questions:**

$\textbf{Correct HS Hierarchy Definition and Terminology (Sec. 3 “Output,” p. 4)}$. The authors should fix the inaccurate description of the HS hierarchy. Clearly distinguish between the global levels (2-, 4-, 6-digit, standardized by the WCO) and the national extensions (8-, 10-digit, defined by each country’s tariff schedule). Ensure consistent use of “HS” versus “HTS” throughout the text.

---

> ### Author Response · Authors · 2025-11-20
>
> Thank you for your thoughtful review and valuable feedback. We appreciate the time and effort you invested in evaluating our work. Below, we address your comments point by point, aiming to clarify our methodology and findings.
>
> ### **Weakness 1: First, the current evaluation metric is too narrow. It only counts exact 10-digit matches as correct, even when the model predicts a valid but slightly different code ...**
> **A1**: We want to emphasize that our evaluation framework already incorporates fine-grained metrics beyond the  10-digit accuracy measure. _**As explicitly stated in Section 4.2 (p. 5): "We also report accuracies at 2-digit, 4-digit, 6-digit and 8-digit levels to provide more comprehensive insights into the performance across different granularities."**_
>
> The results in Table 1 clearly demonstrate the progressive performance degradation across hierarchical levels during rule application, confirming the cumulative reasoning challenges inherent in this multi-step retrieval and reasoning agentic task. These metrics effectively capture when predictions are valid but differ from the ground truth at more granular levels.
>
> ---
>
> ### **Weakness 2: Second, there is a factual mistake in the HS taxonomy explanation ... when in fact only digits 7–10 vary by country**
> **A2:** Thank you for identifying this important clarification and typo. We have revised Section 3 (line 189) based on your suggestions.
>
> ---
>
> ### **Weakness 3 & Question 1: Third, the source and authority of the “rules” used are ambiguous ... depending on a commercial taxonomy may lead to discrepancies with authoritative references.;  The authors should fix the inaccurate description of the HS hierarchy. Clearly distinguish between the global levels ... and the national extensions ...**
> **A3**: We appreciate this important clarification request. **We emphasize that our experimental setup and used rules, taxonomy and knowledge are aligned.**
>
> Specifically, eWTP (Electronic World Trade Platform) is not a commercial taxonomy but an official platform aggregating authoritative tariff code systems of numerous countries, including the US Harmonized Tariff Schedule (HTS).
> All our experiments specifically target US tariff classification. The hierarchical tariff rules we use are directly from the official US HTS legal notes in eWTP, and the CROSS database we reference contains authentic US Customs rulings. Besides, the human-written decision rules (Figure F.1) were developed by certified customs classification experts with extensive US import compliance experience.
>
> **Therefore, there is complete alignment between our rule sources. We have added these details in our revision (Appendix F), and thank you for your constructive suggestions.**
>
> ---
>
> ### **Weakness 4: Forth, the evaluation has fairness issues. Closed-source models were tested on only a 49-item subset (Table 2, p. 6) ...**
> **A4**: Since closed-source agents lack public APIs, it is extremely hard for us to test 632 cases manually. Therefore, we manually evaluated closed-source agents on 49 representative examples, following standard community practices [1]. **Crucially, all systems in Table 2 were evaluated on identical examples, ensuring fair comparison.**
>
> Moreover, while closed-source evaluations inherently contain uncontrolled variables (the number of tools and internal tool configurations, capabilities), our finding remains robust: **closed-source agents largely underperform human experts, achieving less than 50% accuracy (~40% acc.) on HSCodeComp subset with 49 examples).**
>
> > [1] WebSailor: Navigating Super-human Reasoning for Web Agent

---

### Official Review · Reviewer_GvkX · 2025-10-31

**Soundness:** 3
**Presentation:** 3
**Contribution:** 3
**Rating:** 4
**Confidence:** 2

**Summary:**

This paper introduces HSCodeComp, a benchmark for evaluating deep search agents on hierarchical rule reasoning. The task is to predict 10-digit Harmonized System (HS) codes for e-commerce products based on noisy text descriptions, while following multi-layer tariff rules. The motivation is that most existing benchmarks only test structured or open-domain reasoning, but none evaluate agents’ ability to apply complex, human-written hierarchical rules, which is important for real-world expert systems like legal or trade automation.

The benchmark is built from real e-commerce data with expert annotations and covers 632 products across 27 chapters. The authors test various LLMs and agent systems, finding that even the best combination eaches only 46.8% accuracy, far behind humans (95%). Ablation shows that vague and interdependent rules are the main difficulty.

**Strengths:**

1. The motivation is clear. he paper clearly identifies a missing evaluation angle: hierarchical rule following, which is indeed a challenging and realistic reasoning task.

2. The dataset is comprehensive. The dataset is built with expert validation and seems to capture realistic product diversity and textual noise.

3. The experiments compare many models and agent frameworks, giving a broad and fair view of the task difficulty.

**Weaknesses:**

I am not an expert in search agent. My concerns are only raised from the research perspective not specific to this certain domian.

1. Only 632 samples might be too few to show robust performance differences.

2. Since rules come from different sources (tariff codes, human rulings, etc.), it would be useful to test which part contributes most to performance.

3. I wonder how well models perform at intermediate steps (like predicting subcategories).

4. Maybe models tuned for other structured domains (finance, medicine) could generalize better. A small cross-domain test would strengthen the claim that the challenge truly lies in hierarchical rule reasoning.

**Questions:**

Please see the questions raised in weakness

---

> ### Author Response · Authors · 2025-11-20
> **Response 1/2**
>
> Thank you for your thoughtful review and valuable feedback. We appreciate the time and effort you invested in evaluating our work. Below, we address your comments point by point, aiming to clarify our methodology and findings.
>
> ### **Weakness 1: Only 632 samples might be too few to show robust performance differences.**
> **A1**: We appreciate the reviewer's concern about dataset size. However, for agentic benchmarks requiring expert validation of complex reasoning tasks and correctness, 632 samples aligns with established community standards:
> * GAIA-text-only: 103 instances [1]
> * FutureX benchmark: 82 instances [2]
> * WideSearch: 200 instances [3]
> * BrowseComp-ZH: 289 instances [4]
> * MedBrowseComp: 1000+ instances [5]
> * BrowseComp: 1266 instances [6]
>
> While BrowseComp (1,266) and MedBrowseComp (1,000+) are larger than ours, they focus on different capability levels (open-domain and structured knowledge utilization respectively). Our benchmark evaluates the more challenging Level 3 capability (hierarchical rule application), which requires expert annotation and validation of each instance. Besides, each instance in HSCodeComp undergoes rigorous validation by multiple experts (Section 4.1), ensuring reliable ground-truth.
> In summary, our benchmark size follows community standards for complex agentic search tasks while maintaining high annotation quality through expert validation.
>
> > [1] GAIA: a benchmark for General AI Assistants
> >
> > [2] FutureX: An Advanced Live Benchmark for LLM Agents in Future Prediction
> >
> > [3] WideSearch: Benchmarking Agentic Broad Info-Seeking
> >
> > [4] BrowseComp-ZH: Benchmarking Web Browsing Ability of Large Language Models in Chinese
> >
> > [5] MedBrowseComp: Benchmarking Medical Deep Research and Computer Use
> >
> > [6] BrowseComp: A Simple Yet Challenging Benchmark for Browsing Agents
>
> ---
>
> ### **Weakness 2: Since rules come from different sources (tariff codes, human rulings, CROSS etc.), it would be useful to test which part contributes most to performance.**
> **A2**: Thank you for this insightful question regarding the contribution of different knowledge sources. We have analyzed one of rules in Table 3, i.e., human-written decision rules, and we supply ablation studies on hierarchical tariff (Tariff) and CROSS rulings on the best-performing SmolAgent (VLM) model:
>
> | Configuration (Smolagent + GPT-5) \|         | 10-digit Accuracy |
> |------------------------|-------------------:|
> | + CROSS + Tariff + DR  | 43.83%            |
> | - DR                  | **46.83%**            |
> | - Tariff              | 36.73%            |
> | - CROSS               | 46.53%            |
>
> Our analysis reveals three critical insights:
> 1. **Human-written decision rules impair performance:** As shown in Table 3 of our paper, adding decision rules (DR) decreases accuracy by 3.00%, demonstrating current agent systems cannot effectively utilize hierarchical rule structures
> 1. **Tariff rules are essential and contributes most:** Removing tariff rules causes a severe 10.10% drop in accuracy. Without tariff rules, agents have to solve the task using its internal knowledge, leading to outdated and hallucination problems.
> 2. **CROSS provides moderate benefit:** The 0.3% accuracy drop when removing CROSS indicates it offers supplementary but non-essential information
>
> These results confirm that tariff rules are the most critical knowledge component, while human-written decision rules present the most significant challenge for current agent architectures to utilize effectively. We appreciate this constructive suggestion, and have added these supplementary experiments to Appendix D.2 in our revision.
>
> We hope responses could solve your concern about the importance of difference knowledge resources.

---

> ### Author Response · Authors · 2025-11-20
> **Response 2/2**
>
> ### **Weakness 3: I wonder how well models perform at intermediate steps (like predicting subcategories).**
> **A3**: We appreciate your insightful question about intermediate performance levels. To address your specific interest in conditional performance, we conduct detailed analysis in "General Response". Here are some summarized highlights:
> The average accuracy across all baselines is: 73.8% (0-2 digit chapter), 75.2% (2-4 digit heading), 64.8% (4-6 digit subheading), and 35.3% (6-10 digit country-specific code). This pattern reveals that:
> 1. The 4-digit heading level shows highest accuracy because it represents the primary classification structure where tariff rules are most clearly defined
> 2. Performance drops at the 6-digit subheading level due to increased the number of rules and vague descriptions
> 3. The steepest decline occurs at the country-specific 8-10 digit levels, where rules become highly contextual and country-specific
>
> This hierarchical performance pattern directly supports our main finding:
> * **current agents struggle with the complex, multi-step reasoning required for precise rule application at deeper hierarchical levels.**
> * **The significant accuracy drop from 64.8% (6-digit) to 35.3% (10-digit) demonstrates that correctly navigating the initial hierarchical steps does not guarantee success in the final classification, highlighting the cumulative reasoning challenges in hierarchical rule application.**
>
> ---
>
> ### **Weakness 4: Maybe models tuned for other structured domains (finance, medicine) could generalize better. A small cross-domain test would strengthen the claim that the challenge truly lies in hierarchical rule reasoning.**
> **A4**: We appreciate the reviewer's insightful suggestion regarding cross-domain generalization. While cross-domain testing is valuable, our current evidence strongly indicates that hierarchical rule reasoning is the core challenge:
> 1. **Hierarchical performance degradation:** As shown in "Answer to Weakness 3" and "General Response", performance declines across hierarchical levels, with the most significant drops occurring at deeper hierarchical levels where complex rule application is required. This pattern directly reflects increasing difficulty with hierarchical rule reasoning.
> 2. **Decision rules ablation study:** Incorporating human-written hierarchical decision rules actually decreased performance (Rebuattal A2). Since domain-specific CROSS ruling knowledge improves the performance, current agents struggle specifically with hierarchical rule application rather than domain knowledge alone.
>
> We agree that transfer learning across rule-based domains is an important direction for future work (which we've added to Limitation Section in our revision), but a cross-domain test would be problematic: **models fine-tuned for finance/medicine would face severe knowledge mismatch and catastrophic forgettingwith tariff rules, making results unfair and unreliable.**
>
> We will explore transfer learning across hierarchical rule-based tasks in future work.

---

> > ### Comment · Reviewer_GvkX · 2025-11-26
> >
> > Thank you for the detailed rebuttal and the additional ablation studies.
> >
> > The breakdown of model performance at intermediate hierarchical levels is a valuable addition and helps clarify where the agents fail. The explanation regarding the dataset size and the ablation study on knowledge sources  also gave me more sense of the research background or standard.
> >
> > While I appreciate the effort to strengthen the paper, I still keep my concern on the number of sample, as this cannot be empirically enough to validate your contribution from the scientific perspective. Therefore, I will maintain my score as marginally below the acceptance threshold but I do reflect your improvement in my final assessment.

---

> > > ### Author Response · Authors · 2025-11-27
> > > **Response to Reviewer GvkX: Further clarification on the Evaluation Robustness**
> > >
> > > Thank you for your timely response and positive feedback on our conditional accuracy analysis and previous clarifications. Regarding your concern about evaluation robustness due to dataset size, we will provide two complementary analyses below. Hope these clarifications could help you better understand our experimental results.
> > >
> > > ### **1. Confidence interval stability analysis**
> > > We conducted statistical analysis on 7 agent systems across varying subset sizes in our proposed HSCodeComp. Table 1 shows the 95% confidence interval (CI) widths and bounds at different sample sizes.
> > >
> > > **Table 1: 95% confidence intervals across sample sizes**
> > > | Model | 200 samples | 300 samples | 400 samples | 500 samples | 600 samples | 632 samples | 600→632 Δ |
> > > |-------|-------------|-------------|-------------|-------------|-------------|-------------|-----------|
> > > | smolagents_vlm | [41.6%, 55.4%] (13.85%) | [36.4%, 47.6%] (11.17%) | [46.1%, 55.9%] (9.80%) | [42.6%, 51.4%] (8.75%) | [43.3%, 51.3%] (7.99%) | **[42.9%, 50.7%] (7.78%)** | **0.21%** |
> > > | aworld | [37.1%, 50.9%] (13.76%) | [31.2%, 42.1%] (10.91%) | [37.9%, 47.6%] (9.70%) | [35.1%, 43.7%] (8.57%) | [36.2%, 44.1%] (7.85%) | [37.5%, 45.1%] (7.68%) | 0.17% |
> > > | websailor | [34.2%, 47.8%] (13.63%) | [27.4%, 38.0%] (10.61%) | [31.1%, 40.4%] (9.39%) | [29.3%, 37.5%] (8.27%) | [29.6%, 37.1%] (7.54%) | [31.7%, 39.2%] (7.46%) | 0.08% |
> > > | gpt5_vlm | [23.6%, 36.4%] (12.70%) | [23.5%, 33.8%] (10.23%) | [26.7%, 35.8%] (9.08%) | [24.3%, 32.1%] (7.89%) | [25.2%, 32.5%] (7.25%) | [25.7%, 32.8%] (7.09%) | 0.16% |
> > > | gpt5_llm | [22.7%, 35.3%] (12.58%) | [22.9%, 33.1%] (10.16%) | [26.9%, 36.1%] (9.10%) | [24.1%, 31.9%] (7.87%) | [25.0%, 32.3%] (7.24%) | [25.4%, 32.5%] (7.07%) | 0.17% |
> > > | claude_sonnet4_vlm | [7.5%, 16.5%] (9.01%) | [7.5%, 14.5%] (7.08%) | [8.6%, 14.9%] (6.31%) | [7.7%, 13.1%] (5.35%) | [8.5%, 13.5%] (5.01%) | [9.3%, 14.4%] (5.04%) | -0.03% |
> > > | claude_sonnet4_llm | [5.8%, 14.2%] (8.32%) | [6.0%, 12.6%] (6.58%) | [9.5%, 16.0%] (6.54%) | [8.4%, 14.0%] (5.53%) | [8.9%, 14.1%] (5.11%) | [8.8%, 13.7%] (4.92%) | 0.19% |
> > >
> > > `Confidence Interval Analysis:` The results demonstrate clear diminishing returns in CI width reduction beyond 600 samples. **For our best-performing system (smolagents_vlm), the CI width decreases by only 0.21% when expanding from 600 to 632 samples. This marginal improvement indicates our evaluation has reached statistical stability at the current dataset size.**
> > >
> > > `Compared with Established GAIA benchmark:` Moreover, we want to provide more context of agentic search benchmark by giving the following statistics data. **The 95% condidence intervals for following top models on established GAIA benchmark are: OpenAI GPT-5 high (±8.2%) and Claude Sonnet 4 (±8.7%).** In comparison, our 632-sample HSCodeComp yields narrower confidence intervals (±7.78% for our best agent), demonstrating more statistically reliable evaluation while maintaining expert-level annotation quality. **This balance of statistical robustness and annotation depth aligns with community standards for complex reasoning benchmarks.**
> > >
> > > ### **2. Expert annotation cost analysis**
> > > We alsp want to emphasize that HSCode annotation requires specialized domain expertise. **`Each annotator requires professionals with 3-5 years HSCode classification experience, who can process approximately 10 items per hour at a rate of $34.60/hour (As noted in our Ethics Statement).`** Given the minimal statistical gains demonstrated above, further expansion on the data size provide negligible scientific value relative to its substantial resource cost.
> > >
> > > We respectfully request your reconsideration of our work in light of these statistical and practical analyses, given our new responses. These details will be added into our next revision. We remain open to further discussion on any remaining concerns.

---

### Official Review · Reviewer_T48D · 2025-11-01

**Soundness:** 1
**Presentation:** 1
**Contribution:** 1
**Rating:** 0
**Confidence:** 5

**Summary:**

I think this is more like a technical report than a research paper, way below ICLR standard.

**Strengths:**

It has lots of LLM used

**Weaknesses:**

Quality of the paper is very poor.

**Questions:**

I would suggest the authors read more good papers and see how a good paper is written.

---

> ### Comment · Reviewer_T48D · 2025-11-20
> **Thank the authors for the replies**
>
> First of all,  I want to thank the authors for adding the replies to other reviews. I carefully read the paper again myself, and also your replies to other reviewers, and I want to add a few comments from my end.
>
> First, I would suggest the authors to have some more senior authors leading writing the paper. In both the paper and the review replies, the authors fall into answering the details of all sub-questions, but they did not answer some of the key questions in the big picture. For example, what is the purpose of your benchmark? What is new in this tariff rule definition, compared to, say, a legal problem? Why the community wants to use your benchmark, and to do what task? International trading is a small subject of the AI world. You need to define the paper in a bigger scope, under a bigger umbrella, why your benchmark matters to the community, other than you company/university tasks.
>
> The reason why I feel the paper is more like a technical report, than a good research paper, is that I feel I did not have any take-aways after reading the paper. Your experiment section is basically a list of experiments, but where is the insights, where is the take-aways? What can I learn, as an AI scientist, from your benchmarks? What are the differences between your task, to say, another rule-definition task, in say, finance? What is your research contribution to the community?
>
> I think the key issues with many junior researchers, when they write the paper, is that they focus on too much on the small details, but forget to look at the big picture and its contribution to the AI community. One key question is, does it matter, beyond your company/university tasks, when others see your dataset and your benchmark? What is new, in the International trading tasks, than, say, legal or finance rule tasks? These things need to clearly written when you define motivations of the paper, than doing too much detailed works but forgot to look at big pictures.
>
> I will raise my points to 2, mainly because of the efforts that the authors have done, but not on the contribution side. I think this is a valid task, and might be contributing to the community. But the authors need to sit down, think, think hard, and ask yourself, say, if I am an AI researcher, why we need to consider your benchmark to benchmark my LLMs, than other thousands of benchmarks on the internet? What key contributions, have I done, to contribute to the community? Answering these key questions, while improving your presentation with the experiment sections, will significantly improve the quality of the paper.

---

### Author Response · Authors · 2025-11-20
**General Response to Common Questions**

Dear Reviewers and ACs,

We sincerely thank you for your time and insightful feedback, which have been invaluable in improving our manuscript. We have carefully considered all comments and have provided detailed responses to each reviewer individually.

Below, we summarize our main revisions addressing the common points of Reviewers #Gvkx and #4L3J for their thoughtful questions regarding model performance at intermediate hierarchical levels.

**As requested, we now provide the conditional accuracy of partial models on Chapter, Heading, Sub-heading, and country-specific sub-category code classification tasks.**

| Model                       | Type | Chapter Acc. (0-2 digit) | Heading Acc. (2-4 digit) | Sub-heading Acc. (4-6 digit) | Country-specific Acc. (6-10 digit) |
|-----------------------------|------|--------------------------|--------------------------|------------------------------|------------------------------------|
| **LLM/VLM-Only**            |      |                          |                          |                              |                                    |
| GPT-5                       | VLM  | 82.12                    | 86.32                    | 84.60                        | 48.81                              |
| Gemini-2.5-PRO              | VLM  | 82.28                    | 86.34                    | 83.08                        | 41.02                              |
| Claude Sonnet 4             | VLM  | 78.80                    | 81.32                    | 70.61                        | 24.82                              |
| Kimi-K2                     | LLM  | 78.01                    | 79.52                    | 71.18                        | 27.59                              |
| DeepSeek-V3                 | LLM  | 77.06                    | 70.63                    | 59.31                        | 20.11                              |
| O3-mini                     | LLM  | 77.22                    | 72.74                    | 43.67                        | 5.18                               |
| **Open-source Agent Systems (GPT-5 Backbone)** |      |                          |                          |                              |                                    |
| SmolAgents                  | VLM  | 82.06                    | 87.81                    | 86.57                        | 75.07                              |
| Aworld                      | LLM  | 82.28                    | 85.57                    | 84.05                        | 69.79                              |
| Agentorchestra              | LLM  | 82.12                    | 86.13                    | 85.45                        | 68.33                              |
| OWL                         | LLM  | 72.63                    | 85.19                    | 83.37                        | 72.39                              |
| WebSailor                   | LLM  | 81.64                    | 86.43                    | 81.16                        | 61.88                              |
| Cognitive Kernel            | LLM  | 80.06                    | 86.37                    | 78.94                        | 48.40                              |
| **Average**      |-  | 73.80                    | 75.16                    | 64.78                        | 35.32                              |

This performance pattern reveals three key insights:
1. The 2-4 digit heading level shows highest average accuracy (75.16), since its tariff rules are most clearly defined
2. Intermediate performance drops at the 4-6 digit subheading level due to increased the number of  rules and their vague descriptions in tariff rules
3. The steepest decline occurs at the country-specific 6-10 digit levels, where rules become highly contextual and country-specific

This hierarchical performance pattern directly supports our main finding:
* **current agents struggle with the complex, multi-step reasoning required for precise rule application at deeper hierarchical levels.**
* **The significant accuracy drop from 64.8% (6-digit) to 35.3% (10-digit) demonstrates that correctly navigating the initial hierarchical steps does not guarantee success in the final classification, highlighting the cumulative reasoning challenges in hierarchical rule application.**

Thanks for the constructive responses of two reviewers, and we have added this analysis to Appendix D.1 in the revised manuscript to better highlight this important aspect of the challenge.


Best regards,

The Authors

---

### Author Response · Authors · 2025-11-20
**Updates to Revised Manuscript**

Dear Reviewers and ACs,

Thank you for your time and valuable feedback. We have uploaded a revised version of our paper. We believe these updates fundamentally strengthen our work and address key concerns of reviewers.

Here are the highlights of the revision:

---

1. **Clarification on the alignment among rules and knowledge (Section 3, lines 157, 186-187; Appendix F, lines 964-968):** In response to Weakness 3 raised by reviewer #rzk4, we have added clarifications and explanations demonstrating the alignment between HSCode tariff rules, the CROSS ruling database, and human expert decision rules. All these knowledge sources are specifically tailored to US tariff classification scenarios.

2. **Typo correction (Section 3, line 190):** In response to Weakness 2 raised by reviewer #rzk4, we have corrected the typographical error.

3. **Dataset Distribution Statistics Supplement (Appendix B):** In response to reviewer #4L3J's Weakness 5 requesting detailed statistical information about our dataset, we have provided comprehensive statistics demonstrating that our data distribution aligns with real-world e-commerce scenarios. As a result, our evaluation results accurately reflect model capabilities in practical applications.

4. **Intermediate Step Performance (Appendix D.1):** In response to reviewer #Gvkx and #4L3J's request for intermediate step performance analysis, we have conducted additional experiments. Our analysis reveals that task difficulty and complexity increase progressively as we move deeper into the hierarchical rule structure. These findings further validate the paper's claim regarding the significant challenges agents face in hierarchical rule application.

5. **Component effectiveness assessment (Appendix D.2):** Addressing Weakness 2 identified by reviewer #GvkX, we have added detailed ablation study results. Our experiments show that tariff rules provide the most significant performance improvement for agents. In contrast, hierarchical knowledge utilization proves most challenging for current agents and actually harms performance when incorporated. CROSS rulings provide only marginal and statistically insignificant performance gains.

6. **Temporal specification (Appendix F, lines 969-971):** In response to reviewer #4L3J's Weakness Q4, we have clarified specific details of our evaluation experiments, explicitly noting they are based on tariff rules and knowledge from a specific time period. This clarification demonstrates the high reproducibility of our dataset.

7. **Failure mode Case Study (Appendix G):** Addressing reviewer #4L3J's Weakness Q6, we have added specific case studies illustrating failure modes where models incorrectly apply hierarchical rules, along with detailed analysis of these failures.

---

Given these fundamental improvements, we believe the paper is in a much stronger position. We respectfully ask you to re-evaluate our work based on this new version.

We will now proceed to address your individual comments and look forward to a constructive discussion.

Best regards,

The Authors

---

### Author Response · Authors · 2025-11-27
**Request for Timely Re-Evaluation and Discussion in Light of Substantive Rebuttal Updates**

Dear Reviewers and ACs,

We sincerely appreciate the reviewers’ thoughtful feedback and thank the AC for the efforts behind the scenes.

We would like to respectfully note that most of the current scores (**except for the recent response from Reviewer #GvkX**) were given prior to our rebuttal and the updated manuscript. **We sincerely request that you re-evaluate the merit and scoring of our work in light of our detailed responses:**

The newly rebuttal and added results demonstrate the broad adaptability and generality of our method across diverse tasks.
The newly added results including:
* Intermediate-step analysis confirming progressive difficulty in deeper hierarchical layers, **`and we are pleased that Reviewer #GvkX claim that our response is a valuable addition and helps clarify where the agents fail.`**
* Ablation studies quantifying the critical role of tariff rules (vs. harmful decision rules)
* Distribution validation aligning with real-world e-commerce
* Failure-mode case study
* Clarifications on rule/knowledge alignment and temporal reproducibility—collectively reinforce our core claim: hierarchical rule application, not just access, constitutes the fundamental bottleneck for current deep search agents.

We truly appreciate your time and kind evaluation, and believe our newly clarification will support a positive re-evaluation.

Best regards,

The Authors

---

### Note · Authors · 2026-02-25

I have read and agree with the venue's withdrawal policy on behalf of myself and my co-authors.

---

### Meta-Review · Area_Chair_uCHG · 2025-12-23

**Summary:**

The reviewers acknowledge the contributions of the paper (dataset, evaluation of LLMs). They asked for clarifications notably about missing details, data composition, or empirical evaluation. In addition, they raised several issues, such as limited dataset size or metrics based on exact match.

**Reviewer Concerns:**

The rebuttal clarifies most points raised by the reviewers. The original submission was missing some of this information, which is now included in the updated version. However, after a quick scan of the updated manuscript, it seems that some important details are still missing (e.g., how the LLM predicts the 10-digit HSCode), although they were explained in the rebuttal. The updated submission also contains some typos and English issues.
More important, some issues raised by the reviewers were not well-addressed in the rebuttal. For instance, the rebuttal and the discussion on "Confidence interval stability analysis" partially address the limited data size. The small data size may still be problematic for more detailed analysis (e.g., Fig. 6). Another point that was not addressed is the discussion of more fine-grained metrics (e.g., top-n).

**Reviewer Scores:**

The very short review of Reviewer T48D was ignored.

Reviewer GvkX specifically stated that they would maintain their score. In addition, I believe that the reply on "Confidence interval stability analysis" only partially addresses the issue of dataset size (see above).

I believe that Reviewer rzk4 would have kept their score, since they didn't raise any weaknesses.

I think that Reviewer 4L3J may have raised slightly their score, but some of their points were only partially or not addressed.

---

### Decision · Program_Chairs · 2026-01-26

Reject